# Enhanced Federated Optimization:
# Adaptive Unbiased Client Sampling with Reduced Variance

**Dun Zeng**                                            *zengdun@std.uestc.edu.cn*
*University of Electronic Science and Technology of China*
*Peng Cheng Lab*

**Zenglin Xu**[*]                                       *zenglinxu@fudan.edu.cn*
*Fudan University*
*Shanghai Academy of AI for Science*

**Yu Pan**                                              *iperryuu@gmail.com*
*Harbin Institute of Technology, Shenzhen*

**Xu Luo**                                              *frank.luox@outlook.com*
*University of Electronic Science and Technology of China*

**Qifan Wang**                                          *wqfcr618@gmail.com*
*Meta AI*

**Xiaoying Tang**                                       *tangxiaoying@cuhk.edu.cn*
*The Chinese University of Hong Kong, Shenzhen*

**Reviewed on OpenReview:** *https://openreview.net/forum?id=Gb4HBGG9re*

## Abstract

Federated Learning (FL) is a distributed learning paradigm to train a global model across multiple devices without collecting local data. In FL, a server typically selects a subset of clients for each training round to optimize resource usage. Central to this process is the technique of unbiased client sampling, which ensures a representative selection of clients. Current methods primarily utilize a random sampling procedure which, despite its effectiveness, achieves suboptimal efficiency owing to the loose upper bound caused by the sampling variance. In this work, by adopting an independent sampling procedure, we propose a federated optimization framework focused on adaptive unbiased client sampling, improving the convergence rate via an online variance reduction strategy. In particular, we present the first adaptive client sampler, K-Vib, employing an independent sampling procedure. K-Vib achieves a linear speed-up on the regret bound $\tilde{\mathcal{O}}\big(N^{\frac{1}{3}}T^{\frac{2}{3}}/K^{\frac{4}{3}}\big)$ within a set communication budget $K$. Empirical studies indicate that K-Vib doubles the speed compared to baseline algorithms, demonstrating significant potential in federated optimization.

## 1 Introduction

This paper studies the prevalent cross-device federated learning (FL) framework, as outlined in Kairouz et al. (2021); Zhang et al. (2024), which optimizes $\boldsymbol{x} \in \mathcal{X} \subseteq \mathbb{R}^d$ to minimize a finite-sum objective:

$$\min_{\boldsymbol{x} \in \mathcal{X}} f(\boldsymbol{x}) := \sum_{i=1}^{N} \boldsymbol{\lambda}_i f_i(\boldsymbol{x}) := \sum_{i=1}^{N} \boldsymbol{\lambda}_i \mathbb{E}_{\xi_i \sim \mathcal{D}_i}[F_i(\boldsymbol{x}, \xi_i)], \qquad (1)$$

---

[*]Corresponding author.

where $N$ denotes the total number of clients, and $\boldsymbol{\lambda}$ denotes the weights of client objective ($\boldsymbol{\lambda}_i \geq 0, \sum_{i=1}^{N} \boldsymbol{\lambda}_i = 1$). The local loss function $f_i : \mathbb{R}^d \to \mathbb{R}$ is intricately linked to the local data distribution $D_i$. It is defined as $f_i(\boldsymbol{x}) = \mathbb{E}_{\xi_i \sim \mathcal{D}_i}[F_i(\boldsymbol{x}, \xi_i)]$, where $\xi_i$ represents a stochastic batch drawn from $D_i$. Federated optimization algorithms, such as FEDAVG (McMahan et al., 2017), are designed to minimize objectives like Equation (1) by alternating between local and global updates in a distributed learning framework. To reduce communication and computational demands in FL (Konečný et al., 2016; Wang et al., 2021; Yang et al., 2022), various client sampling strategies have been developed (Chen et al., 2020; Cho et al., 2020b; Balakrishnan et al., 2022; Wang et al., 2023; Malinovsky et al., 2023; Cho et al., 2023). These strategies are crucial as they decrease the significant variations in data quality and volume across clients (Khan et al., 2021). Thus, efficient client sampling is key to enhancing the performance of federated optimization.

Current sampling methodologies in FL are broadly divided into biased (Cho et al., 2020b; Balakrishnan et al., 2022; Chen & Vikalo, 2023) and unbiased categories (El Hanchi & Stephens, 2020; Wang et al., 2023). Unbiased client sampling holds particular significance as it maintains the consistency of the optimization objective Wang et al. (2023; 2020). Specifically, unlike biased sampling where client weights $\boldsymbol{\lambda}$ are proportional to sampling probabilities, unbiased methods separate these weights from sampling probabilities. This distinction enables unbiased sampling to be integrated effectively with strategies that address data heterogeneity (Zeng et al., 2023c; Wu et al., 2024; Zeng et al., 2024a), promote fairness (Li et al., 2020c;a), and enhance robustness (Li et al., 2021; 2020a). Additionally, unbiased sampling aligns with secure aggregation protocols for confidentiality in FL (Du & Atallah, 2001; Goryczka & Xiong, 2015; Bonawitz et al., 2017). Hence, unbiased client sampling techniques are indispensable for optimizing federated systems.

Therefore, a better understanding of the implications of unbiased sampling in FL could help us to design better algorithms. To this end, we summarize a general form of federated optimization algorithms with unbiased client sampling in Algorithm 1. Despite differences in methodology, the algorithm covers unbiased sampling techniques (Wang et al., 2023; Malinovsky et al., 2023; Cho et al., 2023; Salehi et al., 2017; Borsos et al., 2018; El Hanchi & Stephens, 2020; Zhao et al., 2021b) in the literature. In Algorithm 1, unbiased sampling comprises three primary steps (referring to lines 3, 12, and 14). First, the **Sampling Procedure** generates a set of samples $S^t$ along with their respective probabilities. Second, the **Global Estimation** step creates global estimates for model updates, aiming to approximate the outcomes as if all participants were involved. Finally, the **Adaptive Strategy** adjusts the sampling probabilities based on the incoming information, ensuring dynamic adaptation to changing data conditions.

Typically, unbiased sampling methods in FL are founded on a **random sampling procedure**, which is then refined to improve global estimation and adaptive strategies. However, the exploration of alternative sampling procedures to enhance unbiased sampling has not been thoroughly investigated. Our research shifts focus to the **independent sampling procedure**, a less conventional approach yet viable for FL. We aim to delineate the distinctions between these methodologies as follows.

> ***Random sampling procedure (RSP)*** *means that the server samples clients from a black box without replacement.*

> ***Independent sampling procedure (ISP)*** *means that the server rolls a dice for every client independently to decide whether to include the client.*

Building on the concept of arbitrary sampling (Horváth & Richtárik, 2019; Chen et al., 2020), our study observes that the ISP can enhance the efficiency of estimating full participation outcomes in FL servers, as detailed in Section 3. However, integrating independent sampling into unbiased techniques introduces new constraints, as outlined in Remark 2.1. Addressing this innovatively in Lemma 5.1, our paper studies the effectiveness of general FL algorithms with adaptive unbiased client sampling, particularly emphasizing the utility and implications of the ISP from an optimization standpoint.

**Contributions** This paper presents a comprehensive analysis of the non-convex convergence in FedAvg and its variants. We first establish a novel link between the cumulative variance of global estimates and convergence rates by separating global estimation results from heterogeneity-related factors. Thus to reduce the cumulative variance, we introduce K-Vib, a novel adaptive sampler incorporating the ISP. Given the expected number

---

**Algorithm 1** FedAvg with Unbiased Client Sampler

---

**Require:** Client set $S$, where $|S| = N$, client weights $\boldsymbol{\lambda}$, times $T$, local steps $R$
1: Initialize sample distribution $\boldsymbol{p}^0$ and model $\boldsymbol{x}^0$
2: **for** time $t$ in $[T]$ **do**
3:     Server runs sampling procedure to create $S^t \sim \boldsymbol{p}^t$
4:     Server broadcasts $\boldsymbol{x}^t$ to sampled clients $i \in S^t$
5:     **for** each client $i \in S^t$ in parallel **do**
6:         $\boldsymbol{x}_i^{t,0} = \boldsymbol{x}^t$
7:         **for** local steps $r$ in $[R]$ **do**
8:             $\boldsymbol{x}_i^{t,r} = \boldsymbol{x}_i^{t,r-1} - \eta_l \nabla F_i(\boldsymbol{x}_i^{t,r-1})$
9:         **end for**
10:        Client uploads local updates $\boldsymbol{g}_i^t = \boldsymbol{x}_i^{t,0} - \boldsymbol{x}_i^{t,R}$
11:     **end for**
12:     Server builds estimates $\boldsymbol{d}^t = \sum_{i \in S^t} \boldsymbol{\lambda}_i \boldsymbol{g}_i^t / \boldsymbol{p}_i^t$
13:     Server updates $\boldsymbol{x}^{t+1} = \boldsymbol{x}^t - \eta_g \boldsymbol{d}^t$
14:     Server updates $\boldsymbol{p}^{t+1}$ based on $\{\|\boldsymbol{g}_i^t\|\}_{i \in S^t}$
15: **end for**

---

of clients $K$ to join federated learning at each round, K-Vib notably achieves an expected regret bound of $\tilde{\mathcal{O}}\left(N^{\frac{1}{3}} T^{\frac{2}{3}} / K^{\frac{4}{3}}\right)$, demonstrating a near-linear speed-up over existing bounds $\tilde{\mathcal{O}}\left(N^{\frac{1}{3}} T^{\frac{2}{3}}\right)$ (Borsos et al., 2018) and $\mathcal{O}\left(N^{\frac{1}{3}} T^{\frac{2}{3}}\right)$ (El Hanchi & Stephens, 2020). Empirically, K-Vib shows accelerated convergence on standard federated tasks compared to baseline algorithms.

## 2 Preliminaries

We first introduce previous works on batch sampling (Horváth & Richtárik, 2019) in stochastic optimization and optimal client sampling (Chen et al., 2020) in FL. We made a few modifications to fit our problem setup.

**Remark 2.1** (Constraints on sampling probability). *We define communication budget $K$ as the expected number of sampled clients. And, its value range is from $1$ to $N$. To be consistent, the sampling probability $\boldsymbol{p}$ always satisfies the constraint $\boldsymbol{p}_i^t > 0, \sum_{i=1}^N \boldsymbol{p}_i^t = K, \forall t \in [T]$ in this paper.*

**Definition 2.1** (Unbiasedness of client sampling $S^t$). *For communication round $t \in [T]$, the estimator $\boldsymbol{d}^t$ is related to sampling probability $\boldsymbol{p}^t$ and the sampling procedure $S^t \sim \boldsymbol{p}^t$. We define a client sampling as unbiased if the sampling $S^t$ and estimates $\boldsymbol{d}^t$ satisfy that*

$$\mathbb{E}_{S^t \sim \boldsymbol{p}^t}[\boldsymbol{d}^t] = \mathbb{E}\left[\sum_{i \in S^t} \frac{\boldsymbol{\lambda}_i \boldsymbol{g}_i^t}{\boldsymbol{p}_i^t}\right] = \sum_{i=1}^N \boldsymbol{\lambda}_i \boldsymbol{g}_i^t.$$

*Besides, the variance of estimator $\boldsymbol{d}^t$ can be derived as:*

$$\mathbb{V}(S^t) := \mathbb{E}_{S^t \sim \boldsymbol{p}^t}\left[\left\|\sum_{i \in S^t} \frac{\boldsymbol{\lambda}_i \boldsymbol{g}_i^t}{\boldsymbol{p}_i^t} - \sum_{i=1}^N \boldsymbol{\lambda}_i \boldsymbol{g}_i^t\right\|^2\right], \tag{2}$$

*where $\mathbb{E}[|S^t|] = K$. We omit the terms $\boldsymbol{\lambda}, \boldsymbol{g}^t$ for notational brevity.*

**Optimal unbiased client sampling** Optimal unbiased client sampling should achieve the lowest variance as defined in Equation (2). It is to estimate the global gradient of full-client participation, *i.e.,*minimize the variance of estimator $\boldsymbol{d}^t$. Given a fixed communication budget $K$, the optimum of the global estimator depends on the collaboration of sampling distribution $\boldsymbol{p}^t$ and the corresponding procedure that outputs $S^t$.

In detail, different sampling procedures associated with the sampling distribution $\boldsymbol{p}$ build a different *probability matrix* $\mathbf{P} \in \mathbb{R}^{N \times N}$, with the elements defined as $\mathbf{P}_{ij} := \text{Prob}(\{i, j\} \subseteq S)$. Arbitrary sampling (Horváth &

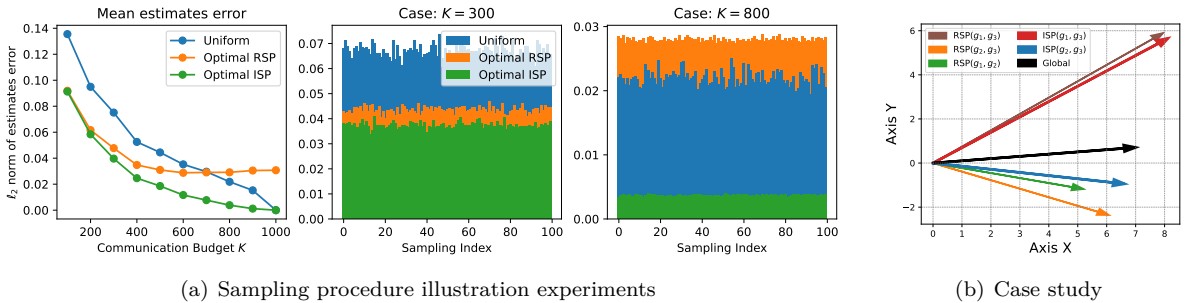

(a) Sampling procedure illustration experiments        (b) Case study

Figure 1: The variance of ISP estimates is lower than RSP. (a) Uniform indicates estimates with uniform probability. (b) The notations RSP($\boldsymbol{g}_i, \boldsymbol{g}_j$) and ISP($\boldsymbol{g}_i, \boldsymbol{g}_j$) represent the global estimates constructed through random sampling and independent sampling, respectively, using sampled vectors $\boldsymbol{g}_i$ and $\boldsymbol{g}_j$. Global indicates the full participation results. We can see ISP($\boldsymbol{g}_i, \boldsymbol{g}_j$) is closer to the Global.

Richtárik, 2019) has shown the generality of denoting arbitrary sampling procedure with a probability matrix for stochastic optimization. Inspired by their findings, we focus on the optimal sampling procedure for the FL server in Lemma 2.1.

**Lemma 2.1** (Optimal sampling procedure, Horváth & Richtárik, 2019)**.** *For any communication round $t \in [T]$ in FL, random sampling yielding the $\mathbf{P}_{ij}^t = Prob(i, j \in S^t) = K(K-1)/N(N-1)$, and independent sampling yielding $\mathbf{P}_{ij}^t = Prob(i, j \in S^t) = \boldsymbol{p}_i^t \boldsymbol{p}_j^t$, they admit*

$$\mathbb{V}(S^t) = \underbrace{\sum_{i=1}^N (1 - \boldsymbol{p}_i^t) \frac{\boldsymbol{\lambda}_i^2 \|\boldsymbol{g}_i^t\|^2}{\boldsymbol{p}_i^t}}_{ISP} \leq \underbrace{\frac{N-K}{N-1} \sum_{i=1}^N \frac{\boldsymbol{\lambda}_i^2 \|\boldsymbol{g}_i^t\|^2}{\boldsymbol{p}_i^t}}_{RSP} . \tag{3}$$

The lemma indicates that the ISP is the optimal sampling procedure that minimizes the upper bound of variance. Then, we have the optimal probability by solving the minimization of the upper bound in respecting probability $\boldsymbol{p}$ in Lemma 2.2.

**Lemma 2.2** (Optimal sampling probability, Chen et al., 2020)**.** *Generally, we can let $\boldsymbol{a}_i = \boldsymbol{\lambda}_i \|\boldsymbol{g}_i^t\|, \forall i \in [N], t \in [T]$ for simplicty of notation. Assuming $0 < \boldsymbol{a}_1 \leq \boldsymbol{a}_2 \leq \cdots \leq \boldsymbol{a}_N$ and $0 < K \leq N$, and $l$ is the largest integer for which $0 < K + l - N \leq \frac{\sum_{i=1}^l \boldsymbol{a}_i}{\boldsymbol{a}_l}$, we have*

$$\boldsymbol{p}_i^* = \begin{cases} (K + l - N) \dfrac{\boldsymbol{a}_i}{\sum_{j=1}^l \boldsymbol{a}_j}, & \text{if } i \leq l, \\ 1, & \text{if } i > l, \end{cases} \quad (ISP) \tag{4}$$

*to be a solution to the optimization problem $\min_{\boldsymbol{p}} \sum_{i=1}^N \frac{\boldsymbol{a}_i^2}{\boldsymbol{p}_i}$. In contrast, we provide the optimal sampling probability for the RSP*

$$\boldsymbol{p}_i^* = K \cdot \frac{\boldsymbol{a}_i}{\sum_{j=1}^N \boldsymbol{a}_j}. \quad (RSP) \tag{5}$$

Therefore, the optimal client sampling in FL uses ISP with probability given in Equation (4).

## 3 Case Study on Sampling Procedure

We suggest designing sampling probability for the ISP to enhance the power of unbiased client sampling in federated optimization. Lemma 2.1 has proven that optimal ISP induces a tighter upper variance bound than optimal RSP. To further clarify it, we provide empirical Example 3.1 and Example 3.2 to demonstrate three superior properties of ISP.

**Example 3.1.** *We randomly generate 1,000 vectors with the size of 1,000 dimensions. We set the budget $K = \{100, 200, \ldots, 1000\}$ and run 100 times the RSP and ISP with their optimal probability from Lemma 2.2 to estimate differences between the output estimates and full aggregation results. Then, we present the mean error of estimate results and two specific cases in Figure 1(a).*

**Example 3.2.** *Consider a case $N = 3, K = 2$ with $\boldsymbol{g}_1 = (\frac{\sqrt{2}}{2}, \frac{\sqrt{2}}{2}), \boldsymbol{g}_2 = (1, -2\sqrt{2}), g_3 = (2\sqrt{7}, 2\sqrt{2})$, it induces weights vector $[\|\boldsymbol{g}_1\|^2, \|\boldsymbol{g}_2\|^2, \|\boldsymbol{g}_3\|^2] = [1, 3, 6]$ if omit $\boldsymbol{\lambda}$. We have optimal sampling probability $\boldsymbol{p}^* = K \cdot [0.1, 0.3, 0.6]$ for RSP and $\boldsymbol{p}^* = [0.25, 0.75, 1]$ for ISP with all possible estimate results in Figure 1(b).*

**ISP handles budget $K$ better than RSP**   With a minimum budget of $K = 1$, the ISP does not assign any client with probability 1, it returns to the optimal sampling probability of RSP by Equation (4). If the budget is $K > 1$, the optimal probability of ISP changes, while the RSP does not. In other words, **The ISP handles the dependency on the budget term $K$ more effectively and produces better estimates compared to RSP**. As illustrated in Figure 1(a), the error in optimal RSP ceases to decrease when $K > 600$. Interestingly, the RSP estimates perform worse than the uniform sampling baseline with a larger budget, as shown in the third plot of Figure 1(a). This discrepancy arises because the optimal probability in RSP aims to minimize a loose upper bound on variance, as defined in equation Equation (3). Specifically, it only minimizes the norm of an estimate $\|\boldsymbol{g}_i/\boldsymbol{p}_i\|^2 - \|\sum \boldsymbol{g}_i/K\|^2$ for each client. Thus, RSP does not account for the precision between its aggregation and true full results.

**ISP estimates are asymptotic to full participation results**   As shown in Figure 1(a), ISP builds estimates asymptotically to the full participation results with an increasing communication budget of $K$, while RSP does not. In Example 3.2, RSP with full participation ($K = 3$) will build estimates $\boldsymbol{d}^t = (6.4, 5.9)$ and non-zero difference $\|\boldsymbol{d}^t - \sum \boldsymbol{g}_i\|^2 = 0.6$. In contrast, ISP with full participation induces $\boldsymbol{p}^* = (1, 1, 1)$ and $\|\boldsymbol{d}^t - \sum \boldsymbol{g}_i\|^2 = 0$. This reveals that RSP may fail to build valid estimates with more communication budget.

**ISP creates expected sampling size**   The number of sampling results from independent sampling is stochastic with expectation $K$. It means that if we strictly conduct the ISP, the number of sampling results $\text{Prob}(|S^t| = K) \neq 1$, but $\mathbb{E}[|S^t|] = K$. Referring to the example, independent sampling may sample 3 clients with probability $p = 0.25 * 0.75 * 1 = 3/16$ and sample only 1 client with $p = (1 - 0.25) * (1 - 0.75) = 3/16$. Importantly, the perturbation of sampling results is acceptable due to the straggler clients (Gu et al., 2021) in a large-scale cross-device FL system. Besides, we can easily extend our analyses to the case with straggler as discussed in Appendix E.1. Moreover, for scenarios that require restricting the probability of ISP from generating a larger selected client set, it induces additional constraints on Remark 2.1. We elaborate on this constraint in Appendix E.2.

The above advantages show that designing a sampling probability and building global estimates with ISP is more promising. However, computing the optimal sampling via Equation (4) requires a norm of full gradients, which is unfeasible in practice. Therefore, FL needs a better design of its sampling probability for ISP based on limited information. In the remainder of this paper, we investigate the efficiency of ISP in federated optimization. **Unless otherwise stated, all sampling probability $\boldsymbol{p}$ and sampling procedures are related to ISP in the remainder of this paper.**

## 4   General Convergence Analyses of FL with Unbiased Client Sampling

In this section, we first provide a general convergence analysis of FedAvg covered by Algorithm 1, specifically focusing on the variance of the global estimator. Our analysis aims to identify the impacts of sampling techniques on enhancing federated optimization. To this end, we define important concepts below to clarify the improvement given by an applied unbiased sampling:

**Definition 4.1** (Sampling quality). *Given communication budget $K$ and arbitrary unbiased client sampling probability $\boldsymbol{p}^t$, we measure the quality (lower is better) of one sampling step $S^t \sim \boldsymbol{p}^t$ by its expectation discrepancy to the optimal sampling:*

$$Q(S^t) := \mathbb{E}_{S^t \sim \boldsymbol{p}^t}\left[\left\|\sum_{i \in S^t} \frac{\boldsymbol{\lambda}_i \boldsymbol{g}_i^t}{\boldsymbol{p}_i^t} - \sum_{i \in S_*^t} \frac{\boldsymbol{\lambda}_i \boldsymbol{g}_i^t}{\boldsymbol{p}_i^*}\right\|^2\right], \tag{6}$$

*where $S_*^t \sim \boldsymbol{p}^*$ is the ISP, $\boldsymbol{p}^*$ is obtained via Equation (4) with full $\{\|\boldsymbol{g}_i^t\|\}_{i\in[N]}$, and $\mathbb{E}[\|S^t\|] = \mathbb{E}[\|S_*^t\|] = K$.*

**Remark**  Note that the second term of Equation (6) denotes the best results that can be possibly obtained subjected to communication budget. It still preserves estimate errors to full results. Therefore, we define the sampling quality of one sampling by its gap to the optimal estimate results for practical concern.

In practical settings, federated learning typically trains modern neural networks, which are non-convex problems in optimization. Therefore, our convergence analyses rely on standard assumptions on the local empirical function $f_i, i \in [N]$ in non-convex federated optimization (Chen et al., 2020; Jhunjhunwala et al., 2022; Chen & Vikalo, 2023).

**Assumption 4.1** (Smoothness)**.** *Each objective $f_i(\boldsymbol{x})$ for all $i \in [N]$ is $L$-smooth, inducing that for all $\forall \boldsymbol{x}, \boldsymbol{y} \in \mathbb{R}^d$, it holds $\|\nabla f_i(\boldsymbol{x}) - \nabla f_i(\boldsymbol{y})\| \le L\|\boldsymbol{x} - \boldsymbol{y}\|$.*

**Assumption 4.2** (Unbiasedness and bounded local variance)**.** *For each $i \in [N]$ and $\boldsymbol{x} \in \mathbb{R}^d$, we assume the access to an unbiased stochastic gradient $\nabla F_i(\boldsymbol{x}, \xi_i)$ of client's true gradient $\nabla f_i(\boldsymbol{x})$, i.e.,$\mathbb{E}_{\xi_i \sim \mathcal{D}_i}[\nabla F_i(\boldsymbol{x}, \xi_i)] = \nabla f_i(\boldsymbol{x})$. The function $f_i$ have $\sigma_l$-bounded (local) variance i.e.,$\mathbb{E}_{\xi_i \sim \mathcal{D}_i}\left[\|\nabla F_i(\boldsymbol{x}, \xi_i) - \nabla f_i(\boldsymbol{x})\|^2\right] \le \sigma_l^2$.*

**Assumption 4.3** (Bounded global variance)**.** *We assume the weight-averaged global variance is bounded, i.e., $\sum_{i=1}^N \boldsymbol{\lambda}_i \|\nabla f_i(\boldsymbol{x}) - \nabla f(\boldsymbol{x})\|^2 \le \sigma_g^2$ for all $\boldsymbol{x} \in \mathbb{R}^d$.*

Assumptions 4.1 and 4.2 are standard assumptions in stochastic optimization analyses. Assumption 4.3 measures the impacts of data heterogeneity in federated optimization. A larger upper bound of $\sigma_g^2$ denotes stronger heterogeneity across clients. Now, we provide the non-convex convergence of Algorithm 1.

**Theorem 4.1** (FedAvg with arbitrary unbiased client sampling)**.** *Under Assumptions 4.1, 4.2, 4.3, taking upper bound $\mathbb{E}\left[f(\boldsymbol{x}^0) - f(\boldsymbol{x}^T)\right] \le M$ and $W = \max\{\boldsymbol{\lambda}_i\}_{i\in[N]}$, given communication budget $K$, there always exists learning rates $\eta_l \eta_g \le \frac{1}{8L}$ allow Algorithm 1 to generate an iteration sequence $\{\boldsymbol{x}^1, \ldots, \boldsymbol{x}^t\}$ such that*

$$\min_{t\in[T]} \mathbb{E}\|\nabla f(\boldsymbol{x}^t)\|^2 \le \underbrace{\mathcal{O}\left(\sqrt{\frac{ML\sigma^2\beta_2}{T}}\right) + \mathcal{O}\left(\sqrt{\frac{M\beta_1}{T}}\right)}_{Sampling\ Utility} + \mathcal{O}\left(\frac{M^{\frac{2}{3}}(\sigma^2)^{\frac{1}{3}}}{T^{\frac{2}{3}}}\right) + \mathcal{O}\left(\frac{ML}{T}\right). \tag{7}$$

*where*

$$\sigma^2 = \Theta(\sigma_l^2/R + \sigma_g^2), \quad \beta_1 = \frac{1}{T}\sum_{t=0}^{T-1} Q(S^t), \quad \beta_2 = \frac{1}{T}\sum_{t=0}^{T-1}\chi^t, \quad \chi^t \in [NW, NW + \frac{N-K}{K}W].$$

*Notably, $\sigma^2$ denotes the heterogeneity impacts on stochastic gradients, $\beta_2$ denotes the benefits of using optimal sampling, and $\beta_1$ denotes the benefits of using sub-optimal sampling.*

**Interpretation of Theorem 4.1**  The convergence guarantees quantify the impacts of client sampling quality on the convergence performance of FedAvg. If we always use optimal client sampling (Chen et al., 2020), the cumulative sampling quality implies $\beta_1 = 0$. Then, the convergence rate returns to Theorem 18 (Chen et al., 2020) and covers the state-of-art complexity guarantees in Karimireddy et al. (2020). As discussed previously, acquiring optimal client sampling is typically unfeasible in FL practice. Therefore, we may use sub-optimal client sampling techniques, which induce $\beta_1$ as a non-zero value.

**Sampling utility**  Given communication budget $K$, the utility of optimal client sampling is linked to $\chi^t \in [NW, NW + \frac{N-K}{K}W]$, where $W = \max\{\boldsymbol{\lambda}_i\}_{i\in[N]}$. For example, $\chi^t = NW$ indicates the best case that optimal client sampling can accurately approximate full results. Otherwise, the optimal client sampling implements sub-optimal approximation. Moreover, given an arbitrary sub-optimal client sampling strategy, we use $Q(S^t)$ to measure the discrepancy between the applied sampling and optimal sampling. Using limited information when optimal client sampling is inapplicable, we expect to minimize the cumulative discrepancies $\beta_1$ to obtain optimization improvement. In all, the bound of the second term in Equation (7) is related to the performance of the applied client sampler in FL. Therefore, minimizing the cumulative sampling quality $\beta_1$ over federated optimization iteration directly accelerates the FL convergence.

# 5 Theories of the K-Vib Sampler

In this section, we introduce the theoretical design of the K-Vib sampler for federated client sampling. The adaptive sampling objective aligns with the online variance reduction (Salehi et al., 2017; Borsos et al., 2018; El Hanchi & Stephens, 2020) tasks in stochastic optimization. The difference is that we solve the problem in the scenario of FL using ISP, which induces the constraints on sampling probability given in Remark 2.1.

## 5.1 Adaptive Client Sampling as Online Optimization

To enhance federated optimization, we aim to minimize the sampling quality $Q(S^t)$ to achieve tighter convergence bound Equation (7). And, using ISP variance, the upper bound of Equation (6) can be known as:

$$Q(S^t) \leq \sum_{i=1}^{N} \frac{\boldsymbol{\lambda}_i^2 \|\boldsymbol{g}_i^t\|^2}{\boldsymbol{p}_i^t} - \sum_{i=1}^{N} \frac{\boldsymbol{\lambda}_i^2 \|\boldsymbol{g}_i^t\|^2}{\boldsymbol{p}_i^*}.$$

Then, we model the client sampling objective as an online convex optimization problem (Salehi et al., 2017; Borsos et al., 2018; El Hanchi & Stephens, 2020). Concretely, we define

$$\text{local feedback function } \pi_t(i) := \boldsymbol{\lambda}_i \|\boldsymbol{g}_i^t\|, \text{ and cost function } \ell_t(\boldsymbol{p}) := \sum_{i=1}^{N} \frac{\pi_t(i)^2}{\boldsymbol{p}_i}$$

for a online convex optimization task[1] respecting sampling probability $\boldsymbol{p}$. Therefore, online convex optimization is to minimize a *dynamic* regret defined as:

$$\frac{1}{T}\sum_{t=1}^{T} Q(S^t) \leq \frac{1}{T}\text{Regret}_D(T) := \frac{1}{T}\left(\sum_{t=1}^{T}\ell_t(\boldsymbol{p}^t) - \sum_{t=1}^{T}\min_{\boldsymbol{p}}\ell_t(\boldsymbol{p})\right). \tag{8}$$

**What does *regret* measure?**   *Regret* measures the cumulative discrepancy of applied sampling probability and the *dynamic* optimal Oracle. In Theorem 4.1, we decomposed the cumulative sampling quality as an error term. And, the upper bound of cumulative sampling quality is given by the *regret*. According to Equation (3), the ISP induces a tighter regret. Minimizing the regret Equation (8) can devise sampling probability for ISP to create a tighter convergence rate for applied FL. In this paper, we are to build an efficient sampler that outputs an exemplary sequence of independent sampling distributions $\{\boldsymbol{p}^t\}_{t=1}^{T}$ such that $\lim_{T\to\infty}\text{Regret}_D(T)/T = 0$.

## 5.2 Analyzing the Best Fixed Probability

In the federated optimization process, the local updates $\boldsymbol{g}^t$ change, making it challenging to directly bound the cumulative discrepancy between the sampling probability and the dynamic optimal probability. Consequently, we explore the advantages of employing the best-fixed probability instead. We decompose the Equation (8) into:

$$\text{Regret}_D(T) = \underbrace{\sum_{t=1}^{T}\ell_t(\boldsymbol{p}^t) - \min_{\boldsymbol{p}}\sum_{t=1}^{T}\ell_t(\boldsymbol{p})}_{\text{Regret}_S(T)} + \underbrace{\min_{\boldsymbol{p}}\sum_{t=1}^{T}\ell_t(\boldsymbol{p}) - \sum_{t=1}^{T}\min_{\boldsymbol{p}}\ell_t(\boldsymbol{p})}_{T_{\text{BFP}}}. \tag{9}$$

The static regret $\text{Regret}_S(T)$ denotes the cumulative online loss gap between an applied sequence of probabilities and the *best-fixed* probability in hindsight. The second term indicates the cumulative loss gap between the best-fixed probability in hindsight and the optimal probabilities. We are to bound the terms respectively.

Our analyses rely on a mild assumption of the convergence status of the federated optimization that sampling methods are applied (Wang et al., 2021). Notably, stochastic optimization (Salehi et al., 2017; Duchi et al.,

---

[1]Please distinguish the online cost function $\ell_t(\cdot)$ from local empirical loss of client $f_i(\cdot)$ and global loss function $f(\cdot)$. While $\ell_t(\cdot)$ is always convex, $f(\cdot)$ and $f_i(\cdot)$ can be non-convex.

---

**Algorithm 2** K-Vib Sampler

---

**Require:** $N$, $K$, $T$, $\gamma$, and $\theta$.

   Initialize client feedback storage $\omega(i) = 0$ for all $i \in [N]$.

   **for** time $t$ in $[T]$ **do**

      $\boldsymbol{p}_i^t \propto \sqrt{\omega(i) + \gamma}$                        $\triangleright$ by Lemma 5.1

      $\tilde{\boldsymbol{p}}_i^t \leftarrow (1 - \theta) \cdot \boldsymbol{p}_i^t + \theta \frac{K}{N}$, for all $i \in [N]$

      Draw $S^t \sim \tilde{\boldsymbol{p}}^t$                            $\triangleright$ ISP

      Receive feedbacks $\pi_t(i)$, and update $\omega(i) \leftarrow \omega(i) + \pi_t^2(i)/\tilde{\boldsymbol{p}}_i^t$ for $i \in S^t$

   **end for**

---

2011; Boyd et al., 2004) and federated optimization algorithms (Reddi et al., 2020; Wang et al., 2020; Li et al., 2019; Zeng et al., 2024b) typically achieve a sub-linear convergence speed $\mathcal{O}(1/\sqrt{T})$ at least. Therefore, we assume feedback function related to the convergence behaviors of local objectives $f_i(\cdot), i \in [N]$ using the following notions:

**Notions** We denote the overall feedback $\Pi_t := \sum_{i=1}^{N} \pi_t(i)$ at $t$-th round. Then, we denote the local convergence results $\pi_*(i) := \lim_{t \to \infty} \pi_t(i)$, and the overall convergence results $\Pi_* := \sum_{i=1}^{N} \pi_*(i)$, $\forall i \in [N]$. Notably, we know $\sum_{t=1}^{T} \Pi_t \geq \Pi_*$ as FL converges and we denote $V_T(i) = \sum_{t=1}^{T} \left( \pi_t(i) - \pi_*(i) \right)^2, \forall T \geq 1,$. Besides, we denote the largest feedback with $G$, i.e., $\pi_t(i) \leq G, \forall t \in [T], i \in [N]$. Importantly, the $G$ denotes the largest feedback during the applied optimization process, instead of assuming bounded gradients.

Then, we assume the convergence of federated optimization will induce a decaying speed of feedback function (local update norms $\|\boldsymbol{g}\|$ in this paper):

**Assumption 5.1** (Convergence of applied federated optimization)**.** *As we discussed above the sub-linear convergence speed $\mathcal{O}(1/\sqrt{T})$ can be obtained by general nonconvex federated learning algorithms. We assume that $|\pi_t(i) - \pi_*(i)| \leq \mathcal{O}(1/\sqrt{t})$, and hence implies $V_T(i) \leq \mathcal{O}(\log(T))$. The above assumptions guarantee the regret concerning a basic convergence speed of applied FL algorithms, with an additional cost of $\tilde{\mathcal{O}}(\sqrt{T})$.*

Now, we bound the second term of Equation (9) below:

**Theorem 5.1** (Bound of best fixed probability)**.** *Under Assumptions 5.1, sampling a set of clients with an expected size of $K$, and for any $i \in [N]$ denote $V_T(i) = \sum_{t=1}^{T} \left( \pi_t(i) - \pi_*(i) \right)^2 \leq \mathcal{O}(\log(T))$. For any $T \geq 1$, the averaged hindsight gap admits,*

$$T_{BFP} \leq \frac{T}{K} \left( \sum_{i=1}^{N} \sqrt{\frac{V_T(i)}{T}} \right) \left( 2\Pi_* + \sum_{i=1}^{N} \sqrt{\frac{V_T(i)}{T}} \right).$$

*Sketch of proof.* This bound can be directly proved to solve the convex optimization problem respectively. Please see Appendix D.1 for details.       $\square$

Theorem 5.1 indicates a fast convergence of federated optimization induces a better bound of $V_T(i)$, yielding a tighter regret. Therefore, it also covers better optimization problems implying a tighter upper bound assumption for $|\pi_t(i) - \pi_*(i)|$ (e.g., strongly convex and convex federated learning problems). As the hindsight bound vanishes with an appropriate FL solver, our objective turns to devise a $\{\boldsymbol{p}_1, \ldots, \boldsymbol{p}_T\}$ that bounds the static regret $\text{Regret}_S(T)$ in Equation (9).

### 5.3 Upper Bound of Static Regret

We utilize the classic follow-the-regularized-leader (FTRL) (Shalev-Shwartz et al., 2012; Kalai & Vempala, 2005; Hazan, 2012) framework to design a stable sampling probability sequence, which is formed at time $t$:

$$\boldsymbol{p}^t := \arg\min_{\boldsymbol{p}} \left\{ \sum_{i=1}^{N} \frac{\pi_{1:t-1}^2(i) + \gamma}{\boldsymbol{p}_i} \right\}, \tag{10}$$

where the regularizer $\gamma$ ensures that the probability does not change too much and prevents assigning a vanishing probability to clients. It also ensures a minimum sampling probability $p_{\min}$ for some clients. Therefore, we have the closed-form solution as shown below:

**Lemma 5.1** (Solution to Equation (10)). *Letting $\boldsymbol{a}_i^t = \pi_{1:t-1}^2(i) + \gamma$ and $0 < \boldsymbol{a}_1^t \leq \boldsymbol{a}_2^t \leq \cdots \leq \boldsymbol{a}_N^t$ and $0 < K \leq N$, we have*

$$\boldsymbol{p}_i^t = \begin{cases} 1, & \text{if } i \geq l_2, \\ z_t \frac{\sqrt{\boldsymbol{a}_i^t}}{c_t}, & \text{if } i \in (l_1, l_2), \\ p_{min}, & \text{if } i \leq l_1, \end{cases} \tag{11}$$

*where $c_t = \sum_{i \in (l_1, l_2)} \sqrt{\boldsymbol{a}_i^t}$, $z_t = K - (N - l_2) + l_1 \cdot p_{min}$ and the $1 \leq l_1 \leq l_2 \leq N$, which satisfies that $\forall i \in (l_1, l_2)$,*

$$\frac{p_{min} \cdot \sum_{l_1 < i < l_2} \boldsymbol{a}_i^t}{z_t} < \boldsymbol{a}_i^t < \frac{\sum_{l_1 < i < l_2} \boldsymbol{a}_i^t}{z_t}.$$

**Remark.** Compared with vanilla optimal sampling probability in Equation (4), our sampling probability especially guarantees a minimum sampling probability $p_{\min}$ on the clients with lower feedback. This probability encourages the exploration of the FL system and prevents the case that some clients are never sampled. Besides, the minimum sampling probability $p_{\min}$ is determined by the $\gamma$ and the cumulative feedback from clients during training. For $t = 1, \ldots, T$, if applied sampling probability follows Lemma 5.1 with a proper $\gamma$, we guarantee that $\text{Regret}_S(T)/T \leq \mathcal{O}(1/\sqrt{T})$, as proved in Appendix D.2.

However, under practical constraints, the server can only access past sampled clients' feedback. Hence, Equation (11) can not be computed accurately. Inspired by work(Borsos et al., 2018), we construct an additional estimate of the true feedback for all clients denoted by $\tilde{\boldsymbol{p}}$ and let $S^t \sim \tilde{\boldsymbol{p}}^t$. Concretely, $\tilde{\boldsymbol{p}}$ is mixed by the original estimator $\boldsymbol{p}^t$ with a static distribution. Let $\theta \in [0, 1]$, we have

$$\text{Mixing strategy:} \quad \tilde{\boldsymbol{p}}^t = (1 - \theta)\boldsymbol{p}^t + \theta \frac{K}{N}, \tag{12}$$

where $\tilde{\boldsymbol{p}}^t \geq \theta \frac{K}{N}$, and hence $\tilde{\pi}_t^2(i) \leq \pi_t^2(i) \cdot \frac{N}{\theta K} \leq G^2 \cdot \frac{N}{\theta K}$.

Analogous to regularizer $\gamma$, the mixing strategy guarantees the least probability that any clients be sampled, thereby encouraging exploration. We present the expected regret bound of the sampling with mixed probability and the K-Vib sampler outlined in Algorithm 2 with theoretical guarantee in Theorem 5.2.

**Theorem 5.2** (Static expected regret with partial feedback). *Under Assumptions 5.1, sampling $S^t \sim \tilde{\boldsymbol{p}}^t$ with $\mathbb{E}[|S^t|] = K$ for all $t = 1, \ldots, T$, and letting $\gamma = G^2 \frac{N}{K\theta}, \theta = (\frac{N}{TK})^{1/3}$ with $T \cdot K \geq N$, we obtain the expected regret*

$$\mathbb{E}[\text{Regret}_S(T)] \leq \tilde{\mathcal{O}}\big(N^{\frac{1}{3}} T^{\frac{2}{3}} / K^{\frac{4}{3}}\big), \tag{13}$$

*where $\tilde{\mathcal{O}}$ hides the logarithmic factors.*

*Sketch of proof.* Denoting $\{\pi_t(i)\}_{i \in S^t}$ as partial feedback from sampled points, it incurs

$$\tilde{\pi}_t^2(i) := \frac{\pi_t^2(i)}{\tilde{\boldsymbol{p}}_i^t} \cdot \mathbb{I}_{i \in S^t}, \text{and } \mathbb{E}[\tilde{\pi}_t^2(i) | \tilde{\boldsymbol{p}}_i^t] = \pi_t^2(i), \forall i \in [N].$$

Analogous to Equation (8), we define modified cost functions and their unbiased estimates:

$$\tilde{\ell}_t(\boldsymbol{p}) := \sum_{i=1}^N \frac{\tilde{\pi}_t^2(i)}{\boldsymbol{p}_i}, \text{and } \mathbb{E}[\tilde{\ell}_t(\boldsymbol{p}) | \tilde{\boldsymbol{p}}^t, \ell_t] = \ell_t(\boldsymbol{p}).$$

Relying on the additional estimates, we have the full cumulative feedback in expectation. In detail, we provide regret bound $\text{Regret}_S(T)$ by directly using Lemma 5.1 in Appendix D.2. Analogously, we can extend the mixed sampling probability $\tilde{\boldsymbol{p}}^t$ to derive the expected regret bound $\mathbb{E}[\text{Regret}_S(T)]$ given in Appendix D.3. $\quad\square$

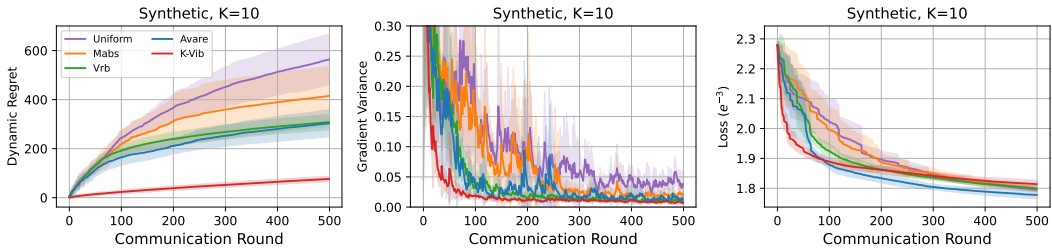

Figure 2: Evaluation on dynamic regre in Equation (8), gradient variance in Equation (2), and loss.

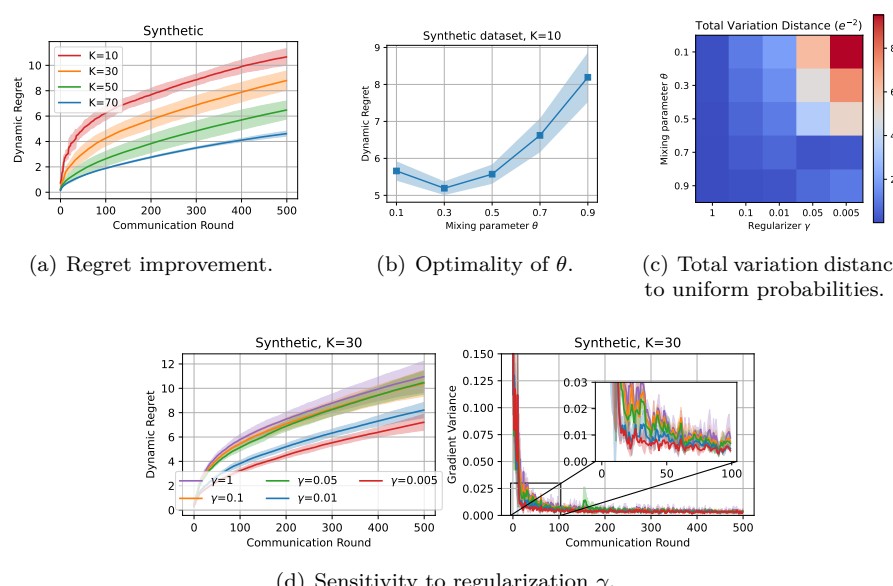

(a) Regret improvement.  (b) Optimality of $\theta$.  (c) Total variation distance to uniform probabilities.

(d) Sensitivity to regularization $\gamma$.

Figure 3: Sensitivity study on synthetic datasets.

**Implications of hyperparameters**  The K-Vib sampler has two key hyperparameters, $\gamma$, $\theta$. The $\gamma$, inherited from the FTRL framework, guarantees the stability of the designed probability sequence. A larger $\gamma$ value limits the extent to which the sampling probability can change after each feedback update. The mixing strategy parameter $\theta$ extends the FTRL framework into the partial feedback scenarios and is tuned to optimize the regret bound. Intuitively, during the early training stages, K-Vib explores the system information using a near-uniform sampling probability, controlled by these two parameters to define the duration of the exploration phase. In the detailed proof in the Appendix, we suggest setting $\gamma = G^2 \frac{N}{\theta K}$ in Equation (50) and $\theta = (\frac{N}{TK})^{\frac{1}{3}}$ in Equation (51) for minimizing expected regret bound.

**Enhanced convergence rate of FedAvg with K-Vib sampler**  The K-Vib sampler can work with a federated optimization process providing unbiased full result estimates. Comparing with previous regret bound $\tilde{\mathcal{O}}\big(N^{\frac{1}{3}}T^{\frac{2}{3}}\big)$ (Borsos et al., 2018) and $\mathcal{O}\big(N^{\frac{1}{3}}T^{\frac{2}{3}}\big)$ (El Hanchi & Stephens, 2020), it implements a linear speed up with communication budget $K$. This advantage relies on a tighter formulation of variance obtained via the ISP. Furthermore, the Assumption 5.1 holds in FedAvg. Meanwhile, the regret bound of the K-Vib sampler is independent of the convergence of Algorithm 1. Therefore, the K-Vib sampler can accelerate FedAvg by minimizing the second term in Theorem 4.1. Concretely, the upper bound of dynamic regret in Equation (9) is dominated by static regret from Theorem 5.2. And then, we combine Equation (8) and substitute $\beta_1$ in Equation (7), which derives that **K-Vib sampler improves the second term** $\mathcal{O}(\sqrt{M\beta_1/T})$ **in Theorem 4.1 to** $\tilde{\mathcal{O}}\left(\frac{M^{1/2}N^{1/6}}{(TK)^{2/3}}\right)$. For computational complexity, the primary cost involves sorting the cumulative feedback sequence $\{\omega(i)\}_{i=1}^N$ in Algorithm 2. This sorting operation can be performed efficiently with an adaptive sorting algorithm (Estivill-Castro & Wood, 1992), resulting in a time complexity of at most $\mathcal{O}(N \log N)$. And, we provide a sketch of efficient implementation in Appendix F.3.

## 6    Experiments

This section evaluates the convergence benefits of utilizing FL client samplers. Our experiment evaluation aligns with previous works (Li et al., 2020b; Chen et al., 2020) on synthetic and Federated EMNIST datasets. And, we additionally evaluate our method on language model and text datasets.

**Baselines**    We compare the K-Vib sampler with the uniform sampling and other adaptive unbiased samplers including Multi-armed Bandit Sampler (Mabs) (Salehi et al., 2017), Variance Reducer Bandit (Vrb) (Borsos et al., 2018) and Avare (El Hanchi & Stephens, 2020). We run experiments with the same random seed and vary the seeds across five independent runs. We present the mean performance with the standard deviation (error bars). To ensure a fair comparison, we set the hyperparameters of all samplers to the optimal values prescribed in the original papers, and concrete hyperparameters are detailed in Appendix F.

**FL and sampler hyperparameters**    We use vanilla SGD optimizers for client-side optimization. And, we fix $\eta_g = 1$ on the server and tune $\eta_l$ for different tasks. For K-Vib sampler, we set $\theta = (\frac{N}{TK})^{\frac{1}{3}}$, which aligns with Theorem 5.2. Then, we guarantee the stability of designed probability via setting $\gamma \approx G^2 \frac{N}{\theta K}$. In practice, we suggest using the mean value of first-round client feedback as a naive estimate of $G$. The first-round feedback is typically the largest during FL training based on Assumption 5.1. This hyperparameter tuning experience can be applied in future applications.

### 6.1    Synthetic Dataset

We evaluate the theoretical results via experiments on Synthetic datasets, where the data are generated from Gaussian distributions (Li et al., 2020b) and the model is logistic regression $f(\boldsymbol{x}) = \arg\max(W^T \boldsymbol{x} + \boldsymbol{b})$. We generate $N = 100$ clients of each has a synthetic dataset, where the size of each dataset follows the power law. We set local learning rate $\eta_l = 0.02$, local epoch 1, and batch size 64.

In Figure 2, we show the action of all samplers on three metrics. Concretely, the K-Vib implements a lower curve of regret in comparison with baselines. Hence, it creates a better estimate with lower variance for global model updating. Connecting with Theorem 4.1, FedAvg with K-Vib achieves a faster convergence.

We present Figure 3(a) to prove the linear speed up about communication budget $K$ in Theorem 5.2. In detail, with the increase of budget $K$, the performance of the K-Vib sampler with regret metric is reduced significantly. Due to page limitation, we provide further illustration examples of other baselines in the same metric in Appendix Figure 7, where we show that the regret bound of baselines methods are not reduced with increasing communication budget $K$. The results demonstrate our unique improvements in theories.

In Figure 3(b), we show the cumulated dynamic regret with $\gamma = 1e^{-4}, T = 500$ and different $\theta = \{0.1, 0.3, 0.5, 0.7, 0.9\}$. The results demonstrate the importance of choosing the proper mixing parameter $\theta$ for regret minimization. And, Figure 3(d) reveals the effects of regularization $\gamma$ in Algorithm 2. The regret slightly changes with different $\gamma$. The variance reduction curves remain stable, indicating the K-Vib sampler is not sensitive to $\gamma$. This is because the regularizer $\gamma$ only decides the minimum probability in solution Equation (11). In Figure 3(c), we show how the hyperparameters control the divergence degree of sampling probabilities from uniform probabilities. Large $\theta$ and small $\gamma$ allow the TV distance to be large.

### 6.2    Federated EMNIST Dataset

We evaluate the proposed sampler on the Federated EMNIST (FEMNIST) following Chen et al., 2020 for image classification. The FEMNIST tasks involve three degrees of unbalanced level (Chen et al., 2020), including FEMNIST v1 (2,231 clients in total, 10% clients hold 82% training images), FEMNIST v2 (1,231 clients in total, 20% client hold 90% training images) and FEMNIST v3 (462 clients in total, 50% client hold 98% training images). We use the same CNN model in (McMahan et al., 2017). We set batch size 20, local epochs 3, $\eta_l = 0.01$, and $K = 111, 62, 23$ as 5% of total clients.

In Figure 4, the variance of data quantity decreased from FEMNIST v1 to FEMNIST v3. We observe that the FedAvg with the K-Vib sampler converges about $3\times$ faster than baseline when achieving 75% accuracy in FEMINIST v1 and $2\times$ faster in FEMINIST v2. At early rounds, the global estimates provided

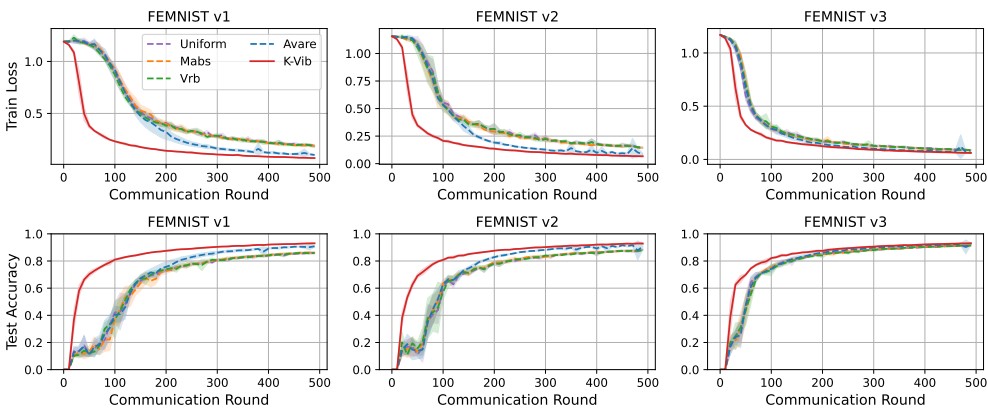

Figure 4: Federated EMNIST dataset experiments. Training loss and test accuracy of FedAvg with different unbiased samplers.

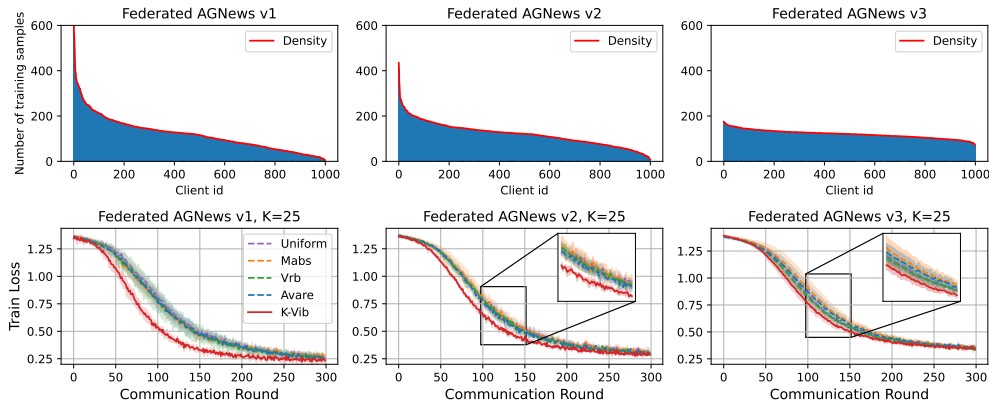

(a) Experiments on AGNews dataset with DistillBert model.

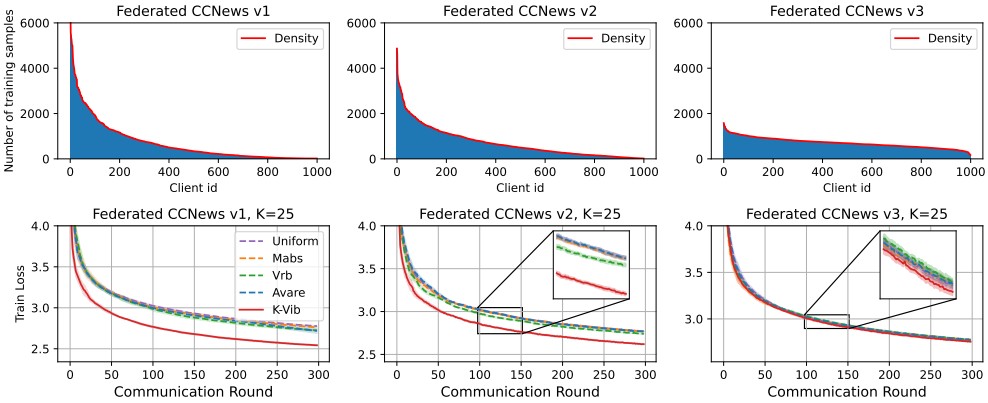

(b) Experiments on CCNews dataset with Pythia-70M model.

Figure 5: Federated text dataset experiments.

by naive independent sampling are better as demonstrated in Lemma 2.1, it induces faster convergence by Theorem 4.1. Meanwhile, the K-Vib sampler further enlarges the convergence benefits by solving an online variance reduction task. Hence, it maintains a fast convergence speed. For baseline methods, we observe that the Vrb and Mabs do not outperform the uniform sampling in the FEMNIST task due to the large number of

clients and large data quantity variance. In contrast, the Avare sampler fastens the convergence curve after about 150 rounds of exploration in the FEMNIST v1 and v2 tasks. On the FEMNIST v3 task, the Avare sampler shows no clear improvement in the convergence curve, while the K-Vib sampler still implements marginal improvements. Horizontally comparing the results, we observe that the curve discrepancy between K-Vib and baselines is the largest in FEMNIST v1. And, the discrepancy narrows with the decrease of data variance across clients. It indicates that the K-Vib sampler works better in the cross-device FL system with a large number of clients and data variance.

### 6.3 CCNews and AGNews Datasets

We evaluate the efficacy of the K-Vib sampler on two large sizes of models and datasets, including a fine-tuning task on AGNews (Zhang et al., 2015) and a pre-training task on CCNews (Mackenzie et al., 2020). AGNews is a text classification task with 119,999 train samples and 4 labels. And, CCNews is a text dataset that contains 708,241 articles. For models, we fine-tune a pretrained language model DistillBert (Sanh et al., 2019) on the AGNews task. DistillBert is a language model with 67 Million parameters. And, we train *from scratch* a GPT2 model called Pythia-70M (Biderman et al., 2023) (70 Million parameters) on CCNews using the next token prediction loss. For both tasks, we partition the datasets into $N = 1,000$ clients with three different levels of heavy long tails (Charles et al., 2024). Then, we set communication round $T = 300$ and budget $K = 25$. We set local learning rate $1e^{-4}$, batch size 16, and epoch 1 for the local SGD optimizer. We report the data distribution and the training loss for comparing the convergence benefits in Figure 5.

Analogous to FEMNIST experiments, we observe that K-Vib achieves $2\times$ faster convergence than baseline methods, while baseline methods only implement a marginal improvement compared to uniform sampling. And, the improvement is related to the degree of data variance across clients. Moreover, we surprisingly observed that baselines can be less efficient than uniform sampling in some cases. This is because RSP derives a loose upper bound, which can be less effective when estimating results of larger dimensions. In the Appendix F.2, we show how the K-Vib sampling probabilities deviate from uniform probabilities. It shows that K-Vib sampling can accurately adapt to local data distribution based on feedback information. In all, our results prove that K-Vib can enhance real-world FL applications, even on large model training.

## 7 Extension, Limitation & Conclusion

Our theoretical findings can be extended to general applications that estimate global results with partial information. Additionally, our extension of independent sampling can be applied to previous works employing random sampling. Besides, the global estimate variance in FL also comes from the data heterogeneity issues, which may incur unstable local feedback, breaking the Assumption 5.1. This can be addressed with client clustering techniques (Ghosh et al., 2020; Ma et al., 2022; Zeng et al., 2023a), analogous to previously cluster sampling works (Fraboni et al., 2021; Song et al., 2023). Besides, we can replace FedAvg with more stable FedAvg variants (Sun et al., 2024; Zeng et al., 2023c).

In conclusion, our study provides a thorough examination of FL frameworks utilizing unbiased client sampling techniques from an optimization standpoint. Our findings highlight the importance of designing unbiased sampling probabilities for the ISP to enhance the efficiency of FL. Building upon this insight, we further extend the range of adaptive sampling techniques and achieve substantial improvements. We are confident that our work will contribute to the advancement of client sampling techniques in FL, making them more applicable and beneficial in various practical scenarios.

### Acknowledgments

We sincerely thank the editors and all anonymous reviewers for their time and constructive comments. This work was partially supported by the Major Key Project of PCL (No. 2022ZD0115301), and the National Key Research and Development Program of China (No. 2018AAA0100204).

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

# Appendices

# A Related Work

Our paper contributes to the literature on the importance sampling in stochastic optimization, online convex optimization, and client sampling in FL.

**Importance Sampling.** Importance sampling is a non-uniform sampling technique widely used in stochastic optimization (Katharopoulos & Fleuret, 2018) and coordinate descent (Richtárik & Takáč, 2016a). Zhao & Zhang (2015); Needell et al. (2014) connects the variance of the gradient estimates and the optimal sampling distribution is proportional to the per-sample gradient norm. The insights of sampling and optimization quality can be transferred into federated client sampling, as we summarised in the following two topics.

**Online Variance Reduction.** Our paper addresses the topic of online convex optimization for reducing variance. Variance reduction techniques are frequently used in conjunction with stochastic optimization algorithms (Defazio et al., 2014; Johnson & Zhang, 2013) to enhance optimization performance. These same variance reduction techniques have also been proposed to quicken federated optimization (Dinh et al., 2020; Malinovsky et al., 2022). On the other hand, online learning (Shalev-Shwartz et al., 2012) typically employs an exploration-exploitation paradigm to develop decision-making strategies that maximize profits. Although some studies have considered client sampling as a multi-armed bandit problem, they have only provided limited theoretical results (Kim et al., 2020; Cho et al., 2020a; Yang et al., 2021). In an intriguing combination, certain studies (Salehi et al., 2017; Borsos et al., 2018; 2019) have formulated data sampling in stochastic optimization as an online learning problem. These methods were also applied to client sampling in FL by treating each client as a data sample in their original problem (Zhao et al., 2021a; El Hanchi & Stephens, 2020).

**Client Sampling in FL.** Client sampling methods in FL fall under two categories: biased and unbiased methods. Unbiased sampling methods ensure objective consistency in FL by yielding the same expected value of results as global aggregation with the participation of all clients. In contrast, biased sampling methods converge to arbitrary sub-optimal outcomes based on the specific sampling strategies utilized. Additional discussion about biased and unbiased sampling methods is provided in Appendix E.3. Recent research has focused on exploring various client sampling strategies for both biased and unbiased methods. For instance, biased sampling methods involve sampling clients with probabilities proportional to their local dataset size (McMahan et al., 2017), selecting clients with a large update norm with higher probability (Chen et al., 2020), choosing clients with higher losses (Cho et al., 2020b), and building a submodular maximization to approximate the full gradients (Balakrishnan et al., 2022). Meanwhile, several studies (Chen et al., 2020; Cho et al., 2020b) have proposed theoretically optimal sampling methods for FL utilizing the unbiased sampling framework, which requires all clients to upload local information before conducting sampling action. Moreover, cluster-based sampling (Fraboni et al., 2021; Xu et al., 2021; Shen et al., 2022) relies on additional clustering operations where the knowledge of utilizing client clustering can be transferred into other client sampling techniques.

# B Useful Lemmas and Corollaries

## B.1 Auxiliary Lemmas

**Lemma B.1** (Lemma 13, Borsos et al., 2018). *For any sequence of numbers $c_1, \ldots, c_T \in [0, 1]$ the following holds:*

$$\sum_{t=1}^{T} \frac{c_t^4}{(c_{1:t}^2)^{3/2}} \leq 44,$$

*where $c_{1:t} = \sum_{\tau=1}^{t} c_\tau$.*

**Lemma B.2.** *For an arbitrary set of $n$ vectors $\{\boldsymbol{a}_i\}_{i=1}^{n}, \boldsymbol{a}_i \in \mathbb{R}^d$,*

$$\left\| \sum_{i=1}^{n} \mathbf{a}_i \right\|^2 \leq n \sum_{i=1}^{n} \|\mathbf{a}_i\|^2. \tag{14}$$

**Lemma B.3.** *For random variables $z_1, \ldots, z_n$, we have*

$$\mathbb{E}\left[\|z_1 + \ldots + z_n\|^2\right] \leq n\mathbb{E}\left[\|z_1\|^2 + \ldots + \|z_n\|^2\right]. \tag{15}$$

**Lemma B.4.** *For independent, mean 0 random variables $z_1, \ldots, z_n$, we have*

$$\mathbb{E}\left[\|z_1 + \ldots + z_n\|^2\right] = \mathbb{E}\left[\|z_1\|^2 + \ldots + \|z_n\|^2\right]. \tag{16}$$

**Lemma B.5** (Tuning the stepsize (Koloskova et al., 2020))**.** *For any parameters $r_0 \geq 0, b \geq 0, e \geq 0, d \geq 0$ there exists constant stepsize $\eta \leq \frac{1}{d}$ such that*

$$\Psi_T := \frac{r_0}{\eta T} + b\eta + e\eta^2 \leq 2\left(\frac{br_0}{T}\right)^{\frac{1}{2}} + 2e^{1/3}\left(\frac{r_0}{T}\right)^{\frac{2}{3}} + \frac{dr_0}{T}.$$

**Lemma B.6** (Upper bound of local drift, Reddi et al., 2020)**.** *Let Assumption 4.2 4.3 hold. For all client $i \in [N]$ with step size $\eta_l \leq \frac{1}{8LR}$ arbitrary local iteration steps $r \in [R]$, the local drift can be bounded as follows,*

$$\mathbb{E}\left\|\boldsymbol{x}_i^{t,r} - \boldsymbol{x}^t\right\|^2 \leq 5R\eta_l^2(\sigma_l^2 + 6R\sigma_g^2 + 6R\left\|\nabla f(\boldsymbol{x}^t)\right\|^2).$$

### B.2 Arbitrary Sampling

In this section, we summarize the arbitrary sampling techniques and present key lemmas used in this paper. The arbitrary sampling is mainly used either for generating mini-batches of samples in stochastic algorithms (Chambolle et al., 2018; Richtárik & Takáč, 2016a) or for coordinate descent optimization (Qu & Richtárik, 2016). In contrast, we explain the background in the context of federated optimization.

In detail, let $S$ denote a sampling, which is a random set-valued mapping with values in $2^{[N]}$, where $[N] := \{1, 2, \ldots, N\}$. An arbitrary sampling $S$ is generated by assigning probabilities to all $2^N$ subsets of $[N]$, which associates a *probability matrix* $\mathbf{P} \in \mathbb{R}^{N \times N}$ defined by

$$\mathbf{P}_{ij} := \text{Prob}(\{i, j\} \subseteq S).$$

Thus, the *probability vector* $p = (\boldsymbol{p}_1, \ldots, \boldsymbol{p}_N) \in \mathbb{R}^N$ is composed of the diagonal entries of $\mathbf{P}$, and $\boldsymbol{p}_i := \text{Prob}(i \in S)$. Furthermore, we say that $S$ is *proper* if $\boldsymbol{p}_i > 0$ for all $i$. Thus, it incurs that

$$K := \mathbb{E}[|S|] = \text{Trace}(\mathbf{P}) = \sum_{i=1}^N \boldsymbol{p}_i.$$

The definition of sampling can be naively transferred to the context of federated client sampling. We refer to $K$ as the expected number of sampled clients per round in FL. The following lemma plays a key role in our problem formulation and analysis.

**Lemma B.7** (Generalization of Lemma 1 Horváth & Richtárik (2019))**.** *Let $\boldsymbol{a}_1, \boldsymbol{a}_2, \ldots, \boldsymbol{a}_N$ be vectors in $\mathbb{R}^d$ and let $\bar{\boldsymbol{a}} = \sum_{i=1}^N \boldsymbol{\lambda}_i \boldsymbol{a}_i$ be their weighted average. Let $S$ be a proper sampling. Assume that there is $\boldsymbol{v} \in \mathbb{R}^N$ such that*

$$\mathbf{P} - pp^t \preceq \boldsymbol{Diag}(\boldsymbol{p}_1\boldsymbol{v}_1, \boldsymbol{p}_2\boldsymbol{v}_2, \ldots, \boldsymbol{p}_N\boldsymbol{v}_N). \tag{17}$$

*Then, we have*

$$\mathbb{E}_{S \sim p}\left[\left\|\sum_{i \in S} \frac{\boldsymbol{\lambda}_i \boldsymbol{a}_i}{\boldsymbol{p}_i} - \bar{\boldsymbol{a}}\right\|^2\right] \leq \sum_{i=1}^N \boldsymbol{\lambda}_i^2 \frac{\boldsymbol{v}_i}{\boldsymbol{p}_i}\|\boldsymbol{a}_i\|^2, \tag{18}$$

*where the expectation is taken over sampling $S$. Whenever Equation (17) holds, it must be the case that*

$$\boldsymbol{v}_i \geq 1 - \boldsymbol{p}_i.$$

*Moreover, The random sampling admits $\boldsymbol{v}_i = \frac{N-K}{N-1}$. The independent sampling admits $\boldsymbol{v}_i = 1 - \boldsymbol{p}_i$ and makes Equation (18) hold as equality.*

*Proof.* Let $\mathbb{I}_{i\in S} = 1$ if $i \in S$ and $\mathbb{I}_{i\in S} = 0$ otherwise. Similarly, let $\mathbb{I}_{i,j\in S} = 1$ if $i \in S$ and $\mathbb{I}_{i,j\in S} = 0$ otherwise. Note that $\mathbb{E}[\mathbb{I}_{i\in S}] = \boldsymbol{p}_i$ and $\mathbb{E}[\mathbb{I}_{i,j\in S}] = \mathbf{P}_{ij}$. Then, we compute the mean of estimates $\tilde{\boldsymbol{a}} := \sum_{i\in S} \frac{\boldsymbol{\lambda}_i \boldsymbol{a}_i}{\boldsymbol{p}_i}$:

$$\mathbb{E}[\tilde{\boldsymbol{a}}] = \mathbb{E}\left[\sum_{i\in S}\frac{\boldsymbol{\lambda}_i\boldsymbol{a}_i}{\boldsymbol{p}_i}\right] = \mathbb{E}\left[\sum_{i=1}^{N}\frac{\boldsymbol{\lambda}_i\boldsymbol{a}_i}{\boldsymbol{p}_i}\mathbb{I}_{i\in S}\right] = \sum_{i=1}^{N}\frac{\boldsymbol{\lambda}_i\boldsymbol{a}_i}{\boldsymbol{p}_i}\mathbb{E}[\mathbb{I}_{i\in S}] = \sum_{i=1}^{N}\boldsymbol{\lambda}_i\boldsymbol{a}_i = \bar{\boldsymbol{a}}.$$

Let $\mathbf{A} = [\boldsymbol{\zeta}_1, \ldots, \boldsymbol{\zeta}_N] \in \mathbb{R}^{d\times N}$, where $\boldsymbol{\zeta}_i = \frac{\boldsymbol{\lambda}_i\boldsymbol{a}_i}{\boldsymbol{p}_i}$, and let $\boldsymbol{e}$ be the vector of all ones in $\mathbb{R}^N$. We now write the variance of $\tilde{\boldsymbol{a}}$ in a form that will be convenient to establish a bound:

$$
\begin{aligned}
\mathbb{E}[\|\tilde{\boldsymbol{a}} - \mathbb{E}[\tilde{\boldsymbol{a}}]\|^2] &= \mathbb{E}[\|\tilde{\boldsymbol{a}}\|^2] - \|\mathbb{E}[\tilde{\boldsymbol{a}}]\|^2 \\
&= \mathbb{E}[\|\sum_{i\in S}\frac{\boldsymbol{\lambda}\boldsymbol{a}_i}{\boldsymbol{p}_i}\|^2] - \|\bar{\boldsymbol{a}}\|^2 \\
&= \mathbb{E}\left[\sum_{i,j}\frac{\boldsymbol{\lambda}_i\boldsymbol{a}_i^\top}{\boldsymbol{p}_i}\frac{\boldsymbol{\lambda}_j\boldsymbol{a}_j}{\boldsymbol{p}_j}\mathbb{I}_{i,j\in S}\right] - \|\bar{\boldsymbol{a}}\|^2 \\
&= \sum_{i,j}\boldsymbol{p}_{ij}\frac{\boldsymbol{\lambda}_i\boldsymbol{a}_i^\top}{\boldsymbol{p}_i}\frac{\boldsymbol{\lambda}_j\boldsymbol{a}_j}{\boldsymbol{p}_j} - \sum_{i,j}\boldsymbol{\lambda}_i\boldsymbol{\lambda}_j\boldsymbol{a}_i^\top\boldsymbol{a}_j \\
&= \sum_{i,j}\left(\boldsymbol{p}_{ij} - \boldsymbol{p}_i\boldsymbol{p}_j\right)\boldsymbol{\zeta}_i^\top\boldsymbol{\zeta}_j \\
&= \boldsymbol{e}^\top\left(\left(\mathbf{P} - \boldsymbol{p}\boldsymbol{p}^\top\right)\circ\mathbf{A}^\top\mathbf{A}\right)\boldsymbol{e}.
\end{aligned}
\tag{19}
$$

Since by assumption we have $\mathbf{P} - \boldsymbol{p}\boldsymbol{p}^\top \preceq \mathbf{Diag}(\boldsymbol{p}\circ\boldsymbol{v})$, we can further bound

$$\boldsymbol{e}^\top\left(\left(\mathbf{P} - \boldsymbol{p}\boldsymbol{p}^\top\right)\circ\mathbf{A}^\top\mathbf{A}\right)\boldsymbol{e} \le \boldsymbol{e}^\top\left(\mathbf{Diag}(\boldsymbol{p}\circ\boldsymbol{v})\circ\mathbf{A}^\top\mathbf{A}\right)\boldsymbol{e} = \sum_{i=1}^{n}\boldsymbol{p}_i\boldsymbol{v}_i\|\boldsymbol{\zeta}_i\|^2. \tag{20}$$

To obtain Equation (18), it remains to combine Equation (20) with Equation (19). Since $\mathbf{P} - \boldsymbol{p}\boldsymbol{p}^\top$ is positive semi-definite (Richtárik & Takáč, 2016b), we can bound $\mathbf{P} - \boldsymbol{p}\boldsymbol{p}^\top \preceq N\mathbf{Diag}(\mathbf{P} - \boldsymbol{p}\boldsymbol{p}^\top) = \mathbf{Diag}(\boldsymbol{p}\circ\boldsymbol{v})$, where $\boldsymbol{v}_i = N(1 - \boldsymbol{p}_i)$.

Overall, arbitrary sampling that associates with a probability matrix $\mathbf{P}$ will determine the value of $\boldsymbol{v}$. As a result, we summarize independent sampling and random sampling as follows,

- Consider now the independent sampling,

$$\mathbf{P} - \boldsymbol{p}\boldsymbol{p}^\top = \begin{bmatrix} \boldsymbol{p}_1(1-\boldsymbol{p}_1) & 0 & \cdots & 0 \\ 0 & \boldsymbol{p}_2(1-\boldsymbol{p}_2) & \cdots & 0 \\ \vdots & \vdots & \ddots & \vdots \\ 0 & 0 & \cdots & \boldsymbol{p}_n(1-\boldsymbol{p}_n) \end{bmatrix} = \mathbf{Diag}(\boldsymbol{p}_1\boldsymbol{v}_1, \ldots, \boldsymbol{p}_n\boldsymbol{v}_n),$$

  where $\boldsymbol{v}_i = 1 - \boldsymbol{p}_i$. Therefore, independent sampling always minimizes Equation (18), making it hold as equality.

- Consider the random sampling,

$$\mathbf{P} - \boldsymbol{p}\boldsymbol{p}^\top = \begin{bmatrix} \frac{K}{N} - \frac{K^2}{N^2} & \frac{K(K-1)}{N(N-1)} & \cdots & \frac{K(K-1)}{N(N-1)} \\ \frac{K(K-1)}{N(N-1)} & \frac{K}{N} & \cdots & \frac{K(K-1)}{N(N-1)} \\ \vdots & \vdots & \ddots & \vdots \\ \frac{K(K-1)}{N(N-1)} & \frac{K(K-1)}{N(N-1)} & \cdots & \frac{K}{N} \end{bmatrix}.$$

As shown in (Horváth & Richtárik, 2019), the standard random sampling admits $\boldsymbol{v}_i = \frac{N-K}{N-1}$ for Equation (18).

$\square$

***Conclusion.*** Given probabilities $\boldsymbol{p}$ that defines all samplings $S$ satisfying $\boldsymbol{p}_i = \text{Prob}(i \in S)$, it turns out that the independent sampling (i.e., $\mathbf{P}_{ij} = \text{Prob}(i, j \in S) = \text{Prob}(i \in S)\text{Prob}(j \in S) = \boldsymbol{p}_i \boldsymbol{p}_j$) minimizes the upper bound in Equation (18). Therefore, depending on the sampling distribution and method, we can rewrite the Equation (18) as follows:

$$\mathbb{V}(S) = \mathbb{E}_{S \sim \boldsymbol{p}}[\| \sum_{i \in S} \frac{\lambda_i \boldsymbol{a}_i}{\boldsymbol{p}_i} - \bar{\boldsymbol{a}}\|^2] = \underbrace{\sum_{i=1}^{N}(1 - \boldsymbol{p}_i)\frac{\lambda_i^2 \|\boldsymbol{a}_i\|^2}{\boldsymbol{p}_i}}_{\text{Independent sampling procedure}} \leq \underbrace{\frac{N-K}{N-1}\sum_{i=1}^{N}\frac{\lambda_i^2 \|\boldsymbol{a}_i\|^2}{\boldsymbol{p}_i}}_{\text{Random sampling procedure}} . \tag{21}$$

## B.3 Proof of Solution to Independent Sampling with Minimal Probability

In this section, we present lemmas and their proofs for our theoretical analyses. Our methodology of independent sampling especially guarantees a minimum probability of clients in comparison with Lemma 2.2. Our proof involves a general constraint, which covers Lemma 2.2. Then, we provide several Corollaries B.1 B.2 for our analysis in the next section.

**Lemma B.8.** *Let $0 < \boldsymbol{a}_1 \leq \boldsymbol{a}_2 \leq \cdots \leq \boldsymbol{a}_N$ and $0 < K \leq N$. We consider the following optimization objective with a restricted probability space $\Delta = \{\boldsymbol{p} \in \mathbb{R}^N | p_{min} \leq \boldsymbol{p}_i \leq 1, \sum_{i=1}^{N} \boldsymbol{p}_i = K, \forall i \in [N]\}$ where $p_{min} \leq K/N$,*

$$\begin{aligned} minimize_{\boldsymbol{p} \in \Delta} \ \Omega(\boldsymbol{p}) &= \sum_{i=1}^{N} \frac{\boldsymbol{a}_i^2}{\boldsymbol{p}_i} \\ subject \ to \ \sum_{i=1}^{N} \boldsymbol{p}_i &= K, \\ p_{min} \leq \boldsymbol{p}_i &\leq 1, \ i = 1, 2, \ldots, N. \end{aligned} \tag{22}$$

*Proof.* We formulate the Lagrangian:

$$\mathcal{L}(p, y, \alpha_1, \ldots, \alpha_N, \beta_1, \ldots, \beta_N) = \sum_{i=1}^{N} \frac{\boldsymbol{a}_i^2}{\boldsymbol{p}_i} + y \cdot \left( \sum_{i=1}^{N} \boldsymbol{p}_i - K \right) + \sum_{i=1}^{N} \alpha_i(p_{\min} - \boldsymbol{p}_i) + \sum_{i=1}^{N} \beta_i(\boldsymbol{p}_i - 1). \tag{23}$$

The constraints are linear and KKT conditions hold. Hence, we have,

$$\boldsymbol{p}_i = \sqrt{\frac{\boldsymbol{a}_i^2}{y - \alpha_i + \beta_i}} = \begin{cases} 1, & \text{if } \sqrt{y} \leq \boldsymbol{a}_i. \\ \sqrt{\frac{\boldsymbol{a}_i^2}{y}}, & \text{if } \sqrt{y} \cdot p_{\min} < \boldsymbol{a}_i < \sqrt{y}, \\ p_{\min}, & \text{if } \boldsymbol{a}_i \leq \sqrt{y} \cdot p_{\min}. \end{cases} \tag{24}$$

Then, we analyze the value of $y$. Letting $l_1 = \left| \{i | \boldsymbol{a}_i \leq \sqrt{y} \cdot p_{\min}\} \right|$, $l_2 = l_1 + |\{\sqrt{y} \cdot p_{\min} < \boldsymbol{a}_i < \sqrt{y}\}|$, $N - l_2 = \left| \{i | \sqrt{y} \leq \boldsymbol{a}_i\} \right|$, and using $\sum_{i=1}^{N} \boldsymbol{p}_i = K$ implies,

$$\sum_{i=1}^{N} \boldsymbol{p}_i = \sum_{i \leq l_1} \boldsymbol{p}_i + \sum_{l_1 < i < l_2} \boldsymbol{p}_i + \sum_{i \geq l_2} \boldsymbol{p}_i = l_1 \cdot p_{\min} + \sum_{l_1 < i < l_2} \sqrt{\frac{\boldsymbol{a}_i^2}{y}} + N - l_2 = K.$$

Arrange the formula, we get

$$\sqrt{y} = \frac{\sum_{l_1 < i < l_2} \boldsymbol{a}_i}{K - N + l_2 - l_1 \cdot p_{\min}}. \tag{25}$$

Moreover, we can plug the results into the objective to get the optimal result:

$$
\begin{aligned}
\sum_{i=1}^{N} \frac{\boldsymbol{a}_i^2}{\boldsymbol{p}_i} &= \sum_{i \leq l_1} \frac{\boldsymbol{a}_i^2}{\boldsymbol{p}_i} + \sum_{l_1 < i < l_2} \frac{\boldsymbol{a}_i^2}{\boldsymbol{p}_i} + \sum_{i \geq N - l_2} \frac{\boldsymbol{a}_i^2}{\boldsymbol{p}_i} \\
&= \frac{\sum_{i \leq l_1} \boldsymbol{a}_i^2}{p_{\min}} + \sqrt{y}\Big( \sum_{l_1 < i < l_2} \boldsymbol{a}_i \Big) + \sum_{i \geq N - l_2} \boldsymbol{a}_i^2 \\
&= \frac{\sum_{i \leq l_1} \boldsymbol{a}_i^2}{p_{\min}} + \frac{(\sum_{l_1 < i < l_2} \boldsymbol{a}_i)^2}{K - N + (l_2 - l_1 \cdot p_{\min})} + \sum_{i \geq N - l_2} \boldsymbol{a}_i^2,
\end{aligned}
\tag{26}
$$

where the $1 \leq l_1 \leq l_2 \leq N$, which satisfies that $\forall i \in (l_1, l_2)$,

$$
p_{\min} \cdot \frac{\sum_{l_1 < i < l_2} \boldsymbol{a}_i}{K - N + l_2 - l_1 \cdot p_{\min}} < \boldsymbol{a}_i < \frac{\sum_{l_1 < i < l_2} \boldsymbol{a}_i}{K - N + l_2 - l_1 \cdot p_{\min}}.
$$

In short, we note that if let $p_{\min} = 0, l_1 = 0$, the Lemma 2.2 is proved as a special case of Equation (26). Besides, we provide further Corollary B.1 and B.2 as preliminaries for further analysis.

**Corollary B.1.** *With $K \cdot a_N \leq \sum_{i=1}^{N} \boldsymbol{a}_i$ and $p_{min} = 0$, we have $l_1 = 0, l_2 = N$ for Equation (26) and induce*

$$
\arg\min \Omega(\boldsymbol{p}^*) = \frac{(\sum_{i=1}^{N} \boldsymbol{a}_i)^2}{K}.
$$

**Corollary B.2.** *With $K \cdot a_N \leq \sum_{i=1}^{N} \boldsymbol{a}_i$ and $p_{min} > 0$, we have $l_2 = N$ and $l_1$ is the largest integer that satisfies $0 < (K - l_1 \cdot p_{min}) \frac{a_{l_1}}{\sum_{i=l_1}^{N} \boldsymbol{a}_i} < p_{min}$. The optimal value of Equation (26) becomes*

$$
\begin{aligned}
\sum_{i=1}^{N} \frac{\boldsymbol{a}_i^2}{\boldsymbol{p}_i} &= \frac{\sum_{i \leq l_1} \boldsymbol{a}_i^2}{p_{min}} + \sqrt{y}\Big( \sum_{l_1 < i \leq N} \boldsymbol{a}_i \Big) && \triangleright \textit{Eq. 26, def. in line 2} \\
&= \frac{\sum_{i \leq l_1} \boldsymbol{a}_i^2}{p_{min}} + y(K - l_1 p_{min}) && \triangleright \textit{Eq. 25, replacing } \sum_{l_1 < i \leq N} \boldsymbol{a}_i \\
&\leq l_1 y p_{min} + y(K - l_1 p_{min}) && \triangleright \textit{Eq. 24, } \boldsymbol{a}_i \leq \sqrt{y} \cdot p_{min} \\
&= \frac{(\sum_{i=l_1}^{N} \boldsymbol{a}_i)^2}{(K - l_1 p_{min})^2} \cdot K \leq \frac{K(\sum_{i=l_1}^{N} \boldsymbol{a}_i)^2}{(K - N p_{min})^2} \\
&\leq \frac{K(\sum_{i=1}^{N} \boldsymbol{a}_i)^2}{(K - N p_{min})^2}.
\end{aligned}
$$

$\square$

## C    Convergence Analyses

### C.1    Analysis on Sampling

We start our convergence analysis with a clarification of the concepts of optimal independent sampling. Considering an Oracle always outputs the optimal probabilities $\boldsymbol{p}^*$, we define

$$
\delta_*^t := \mathbb{E}\left[ \left\| \sum_{i \in S^*} \frac{\boldsymbol{\lambda}_i \boldsymbol{g}_i^t}{\boldsymbol{p}_i^*} - \sum_{i=1}^{N} \boldsymbol{\lambda}_i \boldsymbol{g}_i^t \right\|^2 \right] = \mathbb{E}\left[ \sum_{i=1}^{N} \frac{1 - \boldsymbol{p}_i^*}{\boldsymbol{p}_i^*} \| \tilde{\boldsymbol{g}}_i^t \|^2 \right],
$$

where we have $\|\tilde{\boldsymbol{g}}_i^t\|^2 = \|\boldsymbol{\lambda}_i \boldsymbol{g}_i^t\|^2$. Then, we plug the optimal probability in Equation (4) into the above equation to obtain

$$\delta_*^t = \mathbb{E}\left[\sum_{i=1}^N \frac{1-\boldsymbol{p}_i^*}{\boldsymbol{p}_i^*}\|\tilde{\boldsymbol{g}}_i^t\|^2\right] = \mathbb{E}\left[\frac{1}{K-(N-l)}\left(\sum_{i=1}^l \|\tilde{\boldsymbol{g}}_i^t\|\right)^2 - \sum_{i=1}^l \|\tilde{\boldsymbol{g}}_i^t\|^2\right].$$

Using the fact that $K\|\tilde{\boldsymbol{g}}_N^t\| \leq \sum_{i=1}^N \|\tilde{\boldsymbol{g}}_i^t\|$, we have

$$\delta_*^t \leq \mathbb{E}\left[\frac{1}{K}\left(\sum_{i=1}^N \|\tilde{\boldsymbol{g}}_i^t\|\right)^2 - \sum_{i=1}^N \|\tilde{\boldsymbol{g}}_i^t\|^2\right]$$

$$= \mathbb{E}\left[\frac{1}{K}\left(\sum_{i=1}^N \|\tilde{\boldsymbol{g}}_i^t\|\right)^2\left(1 - K\frac{\sum_{i=1}^N \|\tilde{\boldsymbol{g}}_i^t\|^2}{\left(\sum_{i=1}^N \|\tilde{\boldsymbol{g}}_i^t\|\right)^2}\right)\right]$$

$$\leq \frac{N-K}{NK}\mathbb{E}\left[\left(\sum_{i=1}^N \|\tilde{\boldsymbol{g}}_i^t\|\right)^2\right].$$

To clarify the improvement of utilizing the sampling procedure, we provide two baseline analyses respecting independent sampling and random sampling. For an uniform independent sampling $S^t \sim \mathbb{U}(\boldsymbol{p}_i = \frac{K}{N})$, we have

$$\delta_{\mathbb{U}}^t := \mathbb{E}\left[\left\|\sum_{i\in S^t} \frac{\boldsymbol{\lambda}_i}{\boldsymbol{p}_i}\boldsymbol{g}_i^t - \sum_{i=1}^N \boldsymbol{\lambda}_i \boldsymbol{g}_i^t\right\|^2\right] = \mathbb{E}\left[\sum_{i=1}^N \frac{1-\frac{K}{N}}{\frac{K}{N}}\|\tilde{\boldsymbol{g}}_i^t\|^2\right] = \frac{N-K}{K}\mathbb{E}\left[\sum_{i=1}^N \|\tilde{\boldsymbol{g}}_i^t\|^2\right]. \quad (27)$$

Here, we provide the definition of optimal factor to quantify the improvement of using well-designed sampling probability.

**Definition C.1** (The optimal factor). *Given an iteration sequence of global model $\{\boldsymbol{x}^1, \ldots, \boldsymbol{x}^t\}$, under the constraints of communication budget $K$ and local updates statues $\{\boldsymbol{g}_i^t\}_{i\in[N]}, t \in [T]$, we define the improvement factor of applying optimal client sampling $S_*^t \sim \boldsymbol{p}^*$ comparing uniform sampling $U^t \sim \mathbb{U}$ as:*

$$\alpha_*^t = \frac{\mathbb{E}\left[\left\|\sum_{i\in S_*^t} \frac{\boldsymbol{\lambda}_i}{\boldsymbol{p}_i^*}\boldsymbol{g}_i^t - \sum_{i=1}^N \boldsymbol{\lambda}_i \boldsymbol{g}_i^t\right\|^2\right]}{\mathbb{E}\left[\left\|\sum_{i\in U^t} \frac{\boldsymbol{\lambda}_i}{\boldsymbol{p}_i}\boldsymbol{g}_i^t - \sum_{i=1}^N \boldsymbol{\lambda}_i \boldsymbol{g}_i^t\right\|^2\right]},$$

*and optimal $\boldsymbol{p}^*$ is computed via Equation (4) with $\{\boldsymbol{g}_i^t\}_{i\in[N]}$. Moreover, we can know*

$$\alpha_*^t := \frac{\delta_*^t}{\delta_{\mathbb{U}}} = \frac{\mathbb{E}\left[\left\|\sum_{i\in S^*} \frac{\boldsymbol{\lambda}_i}{\boldsymbol{p}_i^*}\boldsymbol{g}_i^t - \sum_{i=1}^N \boldsymbol{\lambda}_i \boldsymbol{g}_i^t\right\|^2\right]}{\mathbb{E}\left[\left\|\sum_{i\in S^t} \frac{\boldsymbol{\lambda}_i}{\boldsymbol{p}_i}\boldsymbol{g}_i^t - \sum_{i=1}^N \boldsymbol{\lambda}_i \boldsymbol{g}_i^t\right\|^2\right]}$$

$$\leq \frac{K\mathbb{E}\left[\left(\sum_{i=1}^N \|\tilde{\boldsymbol{g}}_i^t\|\right)^2\right]}{NK\mathbb{E}\left[\sum_{i=1}^N \|\tilde{\boldsymbol{g}}_i^t\|^2\right]} < \frac{\mathbb{E}\left[\left(\sum_{i=1}^N \|\tilde{\boldsymbol{g}}_i^t\|\right)^2\right]}{N\mathbb{E}\left[\sum_{i=1}^N \|\tilde{\boldsymbol{g}}_i^t\|^2\right]} \leq 1. \quad (28)$$

## C.2 Non-convex Analyses

Our non-convex convergence analysis follows the standard framework in the optimization literature (Reddi et al., 2020; Chen et al., 2020). In particular, we decompose the sampling quality from the conventional framework in Equation (31), as this paper focuses on the efficiency of sub-optimal client sampling in federated

optimization. Referring to the discussion in the main paper, we can compare the efficiency of the applied sub-optimal samplers in federated optimization and optimal client sampling (Chen et al., 2020).

We start the convergence analysis by recalling the updating rule during round $t$ as:

$$\boldsymbol{x}^{t+1} = \boldsymbol{x}^t - \eta_g \sum_{i \in S^t} \frac{\boldsymbol{\lambda}_i \boldsymbol{g}_i^t}{\boldsymbol{p}_i^t} := \boldsymbol{x}^t - \eta_g \boldsymbol{d}^t, \text{ where } \boldsymbol{g}_i^t = \boldsymbol{x}^t - \boldsymbol{x}_i^{t,R} = \eta_l \sum_{r=1}^R \nabla F_i(\boldsymbol{x}_i^{t,r-1}).$$

Without loss of generality, we rewrite the global descent rule as:

$$\boldsymbol{x}^{t+1} = \boldsymbol{x}^t - \frac{\eta}{R} \sum_{i \in S^t} \frac{\boldsymbol{\lambda}_i}{\boldsymbol{p}_i^t} \tilde{\boldsymbol{g}}_i^t := \boldsymbol{x}^t - \eta \tilde{\boldsymbol{d}}^t,$$

where $\eta = R\eta_l\eta_g$, $\tilde{\boldsymbol{g}}_i^t = \sum_{r=1}^R \nabla F_i(\boldsymbol{x}_i^{t,r-1})$. Therefore, we know $\mathbb{E}_{S^t}[\tilde{\boldsymbol{d}}^t] = \frac{1}{R} \sum_{i=1}^N \boldsymbol{\lambda}_i \tilde{\boldsymbol{g}}_i^t$. Moreover, we denote $W = \max\{\boldsymbol{\lambda}_i\}_{i \in [N]}$.

**Descent lemma.** Using the smoothness of $f$ and taking expectations conditioned on $x^t$ and over the sampling $S^t$, we have

$$
\begin{aligned}
\mathbb{E}\left[f(\boldsymbol{x}^{t+1})\right] &= \mathbb{E}\left[f(\boldsymbol{x}^t - \eta\tilde{\boldsymbol{d}}^t)\right] \leq \mathbb{E}[f(\boldsymbol{x}^t)] - \eta\mathbb{E}[\langle \nabla f(\boldsymbol{x}^t), \mathbb{E}_{S^t}[\tilde{\boldsymbol{d}}^t]\rangle] + \frac{L}{2}\eta^2\mathbb{E}\left[\|\tilde{\boldsymbol{d}}^t\|^2\right] \\
&\leq \mathbb{E}[f(\boldsymbol{x}^t)] - \eta\mathbb{E}\|\nabla f(\boldsymbol{x}^t)\|^2 + \eta\mathbb{E}[\langle\nabla f(\boldsymbol{x}^t), \nabla f(\boldsymbol{x}^t) - \mathbb{E}_{S^t}[\tilde{\boldsymbol{d}}^t]\rangle] + \frac{L}{2}\eta^2\mathbb{E}\left[\|\tilde{\boldsymbol{d}}^t\|^2\right] \\
&\leq f(\boldsymbol{x}^t) - \frac{\eta}{2}\|\nabla f(\boldsymbol{x}^t)\|^2 + \frac{\eta}{2}\mathbb{E}\left[\|\nabla f(\boldsymbol{x}^t) - \mathbb{E}_{S^t}[\tilde{\boldsymbol{d}}^t]\|^2\right] + \frac{L}{2}\eta^2\mathbb{E}\left[\|\tilde{\boldsymbol{d}}^t\|^2\right], \\
&\leq f(\boldsymbol{x}^t) - \frac{\eta}{2}\|\nabla f(\boldsymbol{x}^t)\|^2 + \frac{\eta}{2}\underbrace{\mathbb{E}\left[\|\nabla f(\boldsymbol{x}^t) - \mathbb{E}_{S^t}[\tilde{\boldsymbol{d}}^t]\|^2\right]}_{T_1} + \frac{L}{2}\frac{\eta^2}{R^2\eta_l^2}\underbrace{\mathbb{E}\left[\|\boldsymbol{d}^t\|^2\right]}_{T_2},
\end{aligned}
\tag{29}
$$

where the last inequality follows $\langle a, b\rangle \leq \frac{1}{2}\|a\|^2 + \frac{1}{2}\|b\|^2, \forall a, b \in \mathbb{R}^d$.

**Bounding $T_1$.** We first investigate the expectation gap between global first-order gradient and utilized global estimates,

$$
\begin{aligned}
\mathbb{E}\left[\|\nabla f(\boldsymbol{x}^t) - \mathbb{E}_{S^t}[\tilde{\boldsymbol{d}}^t]\|^2\right] &= \mathbb{E}\left[\left\|\sum_{i=1}^N \boldsymbol{\lambda}_i \nabla f_i(\boldsymbol{x}^t) - \frac{1}{R}\sum_{i=1}^N \boldsymbol{\lambda}_i \tilde{\boldsymbol{g}}_i^t\right\|^2\right] \\
&= \mathbb{E}\left[\left\|\sum_{i=1}^N \boldsymbol{\lambda}_i\left(\nabla f_i(\boldsymbol{x}^t) - \frac{1}{R}\sum_{r=1}^R \nabla F_i(\boldsymbol{x}_i^{t,r-1})\right)\right\|^2\right] \\
&= \mathbb{E}\left[\left\|\sum_{i=1}^N \boldsymbol{\lambda}_i \frac{1}{R}\sum_{r=1}^R\left(\nabla f_i(\boldsymbol{x}^t) - \nabla F_i(\boldsymbol{x}_i^{t,r-1}) + \nabla f_i(\boldsymbol{x}_i^{t,r-1}) - \nabla f_i(\boldsymbol{x}_i^{t,r-1})\right)\right\|^2\right] \\
&\leq 2\mathbb{E}\left[\left\|\sum_{i=1}^N \boldsymbol{\lambda}_i \frac{1}{R}\sum_{r=1}^R\left(\nabla f_i(\boldsymbol{x}^t) - \nabla f_i(\boldsymbol{x}_i^{t,r-1})\right)\right\|^2\right] + 2\mathbb{E}\left[\left\|\sum_{i=1}^N \boldsymbol{\lambda}_i \frac{1}{R}\sum_{r=1}^R\left(\nabla f_i(\boldsymbol{x}_i^{t,r-1}) - \nabla F_i(\boldsymbol{x}_i^{t,r-1})\right)\right\|^2\right]
\end{aligned}
$$

Then, using the L-smoothness of $f$ and unbiasedness of local stochastic gradients, we know

$$
\begin{aligned}
\mathbb{E}\left[\|\nabla f(\boldsymbol{x}^t) - \mathbb{E}_{S^t}[\tilde{\boldsymbol{d}}^t]\|^2\right] &\leq 2\mathbb{E}\left[\left\|\sum_{i=1}^N \boldsymbol{\lambda}_i \frac{1}{R}\sum_{r=1}^R \left(\nabla f_i(\boldsymbol{x}^t) - \nabla f_i(\boldsymbol{x}_i^{t,r-1})\right)\right\|^2\right] + 2\frac{\sigma_l^2}{R} \\
&\leq 2L^2 \sum_{i=1}^N \boldsymbol{\lambda}_i \frac{1}{R}\sum_{r=1}^R \mathbb{E}\left\|\boldsymbol{x}^t - \boldsymbol{x}_i^{t,r-1}\right\|^2 + 2\frac{\sigma_l^2}{R} \\
&\leq 2L^2 \sum_{i=1}^N \boldsymbol{\lambda}_i(5R\eta_l^2\sigma_l^2 + 30R^2\eta_l^2\mathbb{E}\left\|\nabla f_i(\boldsymbol{x}^t)\right\|^2) + 2\frac{\sigma_l^2}{R} \qquad \triangleright \text{ using Lemma B.6} \\
&\leq 10RL^2\eta_l^2\sigma_l^2 + 60R^2L^2\eta_l^2 \sum_{i=1}^N \boldsymbol{\lambda}_i\mathbb{E}\left\|\nabla f_i(\boldsymbol{x}^t)\right\|^2 + 2\frac{\sigma_l^2}{R} \\
&\leq 10RL^2\eta_l^2\sigma_l^2 + 60R^2L^2\eta_l^2(\mathbb{E}\|\nabla f(\boldsymbol{x}^t)\|^2 + \sigma_g^2) + 2\frac{\sigma_l^2}{R} \\
&\leq 60R^2L^2\eta_l^2\mathbb{E}\|\nabla f(\boldsymbol{x}^t)\|^2 + (10RL^2\eta_l^2 + \frac{2}{R})\sigma_l^2 + 60R^2L^2\eta_l^2\sigma_g^2.
\end{aligned}
$$

Then, using $\frac{1}{16RL} \leq \eta_l \leq \frac{1}{8RL}$, we have

$$
\begin{aligned}
\mathbb{E}\left[\|\nabla f(\boldsymbol{x}^t) - \mathbb{E}_{S^t}[\tilde{\boldsymbol{d}}^t]\|^2\right] &\leq \frac{15}{16}\mathbb{E}\|\nabla f(\boldsymbol{x}^t)\|^2 + 10L^2\left((1 + \frac{1}{5\eta_l^2 R^2 L^2})\sigma_l^2 + 6R\sigma_g^2\right)R\eta_l^2 \\
&\leq \frac{15}{16}\mathbb{E}\|\nabla f(\boldsymbol{x}^t)\|^2 + 60L^2\left(9\sigma_l^2/R + \sigma_g^2\right)R^2\eta_l^2
\end{aligned} \tag{30}
$$

**Bounding $T_2$.** Now, we need to bound estimates:

$$
\begin{aligned}
\mathbb{E}\left[\|\boldsymbol{d}^t\|^2\right] &\leq \mathbb{E}\left[\left\|\sum_{i\in S^t}\frac{\boldsymbol{\lambda}_i\boldsymbol{g}_i^t}{\boldsymbol{p}_i^t} - \sum_{i=1}^N \boldsymbol{\lambda}_i\boldsymbol{g}_i^t\right\|^2 + \left\|\sum_{i=1}^N \boldsymbol{\lambda}_i\boldsymbol{g}_i^t\right\|^2\right] \\
&\leq \underbrace{\mathbb{E}\left[\left\|\sum_{i\in S^t}\frac{\boldsymbol{\lambda}_i\boldsymbol{g}_i^t}{\boldsymbol{p}_i^t} - \sum_{i\in S^*}\frac{\boldsymbol{\lambda}_i\boldsymbol{g}_i^t}{\boldsymbol{p}_i^*}\right\|^2\right]}_{Q(S^t)} + \underbrace{\mathbb{E}\left[\left\|\sum_{i\in S^*}\frac{\boldsymbol{\lambda}_i\boldsymbol{g}_i^t}{\boldsymbol{p}_i^*} - \sum_{i=1}^N \boldsymbol{\lambda}_i\boldsymbol{g}_i^t\right\|^2\right] + \mathbb{E}\left[\left\|\sum_{i=1}^N \boldsymbol{\lambda}_i\boldsymbol{g}_i^t\right\|^2\right]}_{(A)}.
\end{aligned} \tag{31}
$$

Here, the $Q(S^t)$ indicates the discrepancy between applied sampling and optimal sampling. The term $(A)$ indicates the intrinsic gap for the optimal sampling to approach its targets and the quality of the targets for optimization.

We can bound the term $(A)$ as follows:

$$(A) = \mathbb{E}\left[\left\|\sum_{i \in S^*} \frac{\boldsymbol{\lambda}_i \boldsymbol{g}_i^t}{\boldsymbol{p}_i^*} - \sum_{i=1}^N \boldsymbol{\lambda}_i \boldsymbol{g}_i^t\right\|^2\right] + \mathbb{E}\left\|\sum_{i=1}^N \boldsymbol{\lambda}_i \boldsymbol{g}_i^t\right\|^2$$

$$\leq \alpha_*^t \frac{N-K}{K} \mathbb{E}\left[\sum_{i=1}^N \boldsymbol{\lambda}_i^2 \left\|\boldsymbol{g}_i^t\right\|^2\right] + \mathbb{E}\left\|\sum_{i=1}^N \boldsymbol{\lambda}_i \boldsymbol{g}_i^t\right\|^2 \qquad \triangleright \text{ using Eq. 27 and Eq. 28}$$

$$\leq \alpha_*^t \frac{N-K}{K} \sum_{i=1}^N \boldsymbol{\lambda}_i^2 \mathbb{E}\left\|\boldsymbol{g}_i^t\right\|^2 + N \sum_{i=1}^N \boldsymbol{\lambda}_i^2 \mathbb{E}\left\|\boldsymbol{g}_i^t\right\|^2$$

$$= \left(\alpha_*^t \frac{N-K}{K} + N\right) \sum_{i=1}^N \boldsymbol{\lambda}_i^2 \mathbb{E}\left\|\boldsymbol{g}_i^t\right\|^2$$

$$\leq \left(\frac{\alpha_*^t(N-K)}{K} + N\right) W \sum_{i=1}^N \boldsymbol{\lambda}_i \mathbb{E}\left\|\boldsymbol{g}_i^t\right\|^2$$

$$\leq \left(\frac{\alpha_*^t(N-K)}{K} + N\right) W \eta_l^2 (5R\sigma_l^2 + 30R^2 \sum_{i=1}^N \boldsymbol{\lambda}_i \mathbb{E}\left\|\nabla f_i(\boldsymbol{x}^t)\right\|^2) \qquad \triangleright \text{ using Lemma B.6}$$

$$\leq \left(\frac{\alpha_*^t(N-K)}{K} + N\right) W \eta_l^2 (5R\sigma_l^2 + 30R^2 (\sigma_g^2 + \mathbb{E}\left\|\nabla f(\boldsymbol{x}^t)\right\|^2)). \qquad \triangleright \text{ using Assumption 4.3}$$

Letting $\chi^t = \left(\frac{\alpha_*^t(N-K)}{K} + N\right) W$, we obtain

$$(A) = 5R\chi^t(\sigma_l^2 + 6R\sigma_g^2)\eta_l^2 + 30R^2\chi^t\eta_l^2 \mathbb{E}\left\|\nabla f(\boldsymbol{x}^t)\right\|^2$$

Therefore, we have

$$T_2 = \mathbb{E}\left[\|\boldsymbol{d}^t\|^2\right] \leq Q(S^t) + 5\chi^t R^2 \eta_l^2 (\sigma_l^2/R + 6\sigma_g^2) + 30\chi^t R^2 \eta_l^2 \mathbb{E}\left\|\nabla f(\boldsymbol{x}^t)\right\|^2 \tag{32}$$

**Putting together.** Reorganizing the descent lemma, we obtain

$$\|\nabla f(\boldsymbol{x}^t)\|^2 \leq \frac{2(f(\boldsymbol{x}^t) - \mathbb{E}\left[f(\boldsymbol{x}^{t+1})\right])}{\eta} + T_1 + \frac{L\eta}{R^2\eta_l^2} \cdot T_2$$

Then, taking full expectation on both sides and substituting corresponding terms in Equation (29) with Equation (30) and Equation (32) to finish the descent lemma, we have

$$\mathbb{E}\|\nabla f(\boldsymbol{x}^t)\|^2 \leq \frac{2(f(\boldsymbol{x}^t) - \mathbb{E}\left[f(\boldsymbol{x}^{t+1})\right])}{\eta}$$

$$+ \frac{15}{16}\mathbb{E}\|\nabla f(\boldsymbol{x}^t)\|^2 + 60L^2\left(9\sigma_l^2/R + \sigma_g^2\right)R^2\eta_l^2$$

$$+ \frac{L\eta}{R^2\eta_l^2}Q(S^t) + \eta \cdot 5\chi^t L(\sigma_l^2/R + 6\sigma_g^2) + \eta \cdot 30L\chi^t \mathbb{E}\left\|\nabla f(\boldsymbol{x}^t)\right\|^2$$

Using $\eta = R\eta_l\eta_g$ and $\frac{1}{16RL} \leq \eta_l \leq \frac{1}{8RL}$, we have

$$\frac{1}{16}(1 - 480L\chi^t\eta)\|\nabla f(\boldsymbol{x}^t)\|^2 \leq \frac{2(f(\boldsymbol{x}^t) - \mathbb{E}\left[f(\boldsymbol{x}^{t+1})\right])}{\eta} + \left(\frac{1}{256L}Q(S^t) + 5\chi^t L\left(\sigma_l^2/R + 6\sigma_g^2\right)\right)\eta$$

$$+ \frac{60L^2}{\eta_g^2}(\sigma_l^2/R + 6\sigma_g^2)\eta^2 \tag{33}$$

$$\leq \frac{2(f(\boldsymbol{x}^t) - \mathbb{E}\left[f(\boldsymbol{x}^{t+1})\right])}{\eta} + \left(\frac{1}{256L}Q(S^t) + 5\chi^t L\sigma^2\right)\eta + \sigma^2\eta^2$$

where the last inequality uses $\sigma^2 = \Theta(\sigma_l^2/R + \sigma_g^2)$ and sets $\eta_g \geq \frac{1}{8L}$ without loss of generality.

Then, we have the descent lemma

$$\frac{(1 - 480L\chi^t\eta)}{16}\|\nabla f(\boldsymbol{x}^t)\|^2 \leq \frac{2(f(\boldsymbol{x}^t) - \mathbb{E}\left[f(\boldsymbol{x}^{t+1})\right])}{\eta} + \left(\frac{1}{256L}Q(S^t) + 5\chi^t L\sigma^2\right)\eta + \sigma^2\eta^2 \qquad (34)$$

Then, taking averaging of both sides of Equation (34) over from time 0 to $T-1$, we have

$$\frac{1}{T}\sum_{t=1}^{T}\frac{(1 - 480L\chi^t\eta)}{16}\mathbb{E}\|\nabla f(\boldsymbol{x}^t)\|^2 \leq \frac{2(\mathbb{E}\left[f(\boldsymbol{x}^0) - f(\boldsymbol{x}^T)\right])}{T\eta} + (\frac{1}{T}\sum_{t=0}^{T-1}Q(S^t)\cdot\frac{1}{256L} + \frac{1}{T}\sum_{t=0}^{T-1}\chi^t\cdot 5L\sigma^2)\eta + \sigma^2\eta^2.$$

Supposing upper bound $\mathbb{E}[f(\boldsymbol{x}^0) - f(\boldsymbol{x}^T)] \leq M$ and defining

$$\beta_1 = \frac{1}{T}\sum_{t=0}^{T-1}Q(S^t), \quad \beta_2 = \frac{1}{T}\sum_{t=0}^{T-1}\chi^t, \quad \rho = \min_{t\in[T]}\{\frac{(1 - 480L\chi^t\eta)}{16}\} > 0$$

we use Lemma B.5 with $\eta \leq \frac{1}{8L}$ to tune the stepsize $\eta$ and obtain

$$\rho\min_{t\in[T]}\mathbb{E}\|\nabla f(\boldsymbol{x}^t)\|^2 \leq 2\left(\frac{2M(\beta_1/256L + 5L\sigma^2\beta_2)}{T}\right)^{\frac{1}{2}} + 2(5\sigma^2)^{\frac{1}{3}}\left(\frac{2M}{T}\right)^{\frac{2}{3}} + \frac{16LM}{T},$$

which concludes the proof.

## D    Detail Proofs of Online Convex Optimization

### D.1    Vanising Hindsight Gap: Proof of Theorem 5.1

We prove this theorem by directly solving a convex optimization problem, w.r.t the sampling probability.

We first arrange the term $T_{BFP}$ in Equation (9) as follows,

$$\min_{\boldsymbol{p}}\sum_{t=1}^{T}\ell_t(\boldsymbol{p}) - \sum_{t=1}^{T}\min_{\boldsymbol{p}}\ell_t(\boldsymbol{p}) = \min_{\boldsymbol{p}}\sum_{t=1}^{T}\sum_{i=1}^{N}\frac{\pi_t^2(i)}{\boldsymbol{p}_i} - \sum_{t=1}^{T}\min_{\boldsymbol{p}}\sum_{i=1}^{N}\frac{\pi_t^2(i)}{\boldsymbol{p}_i}. \qquad (35)$$

Here, we recall our mild Assumption 5.1,

$$\pi_*(i) := \lim_{t\to\infty}\pi_t(i), \ \Pi_* := \sum_{i=1}^{N}\pi_*(i), \ \forall i \in [N].$$

Then, denoting $V_T(i) := \sum_{t=1}^{T}(\pi_t(i) - \pi_*(i))^2$, we bound the cumulative variance over time $T$ per client $i \in [N]$,

$$\pi_{1:T}^2(i) = \sum_{t=1}^{T}(\pi_*(i) + (\pi_t(i) - \pi_*(i)))^2$$

$$\leq T\cdot\pi_*^2(i) + 2\pi_*(i)\sum_{t=1}^{T}|\pi_t(i) - \pi_*(i)| + \sum_{t=1}^{T}(\pi_t(i) - \pi_*(i))^2 \qquad (36)$$

$$\leq T\cdot\pi_*^2(i) + 2\pi_*(i)\sqrt{T\cdot V_T(i)} + V_T(i)$$

$$= T\left(\pi_*(i) + \sqrt{\frac{V_T(i)}{T}}\right)^2.$$

Using the Lemma 2.2 and non-negativity of feedback we have,

$$\min_{\boldsymbol{p}} \sum_{i=1}^{N} \frac{\pi_t^2(i)}{\boldsymbol{p}_i} = \frac{(\sum_{i=1}^{N} \pi_t(i))^2}{K}. \tag{37}$$

We obtain the upper bound of the first term in Equation (35),

$$
\begin{aligned}
\min_{\boldsymbol{p}} \sum_{t=1}^{T} \sum_{i=1}^{N} \frac{\pi_t^2(i)}{\boldsymbol{p}_i} &= \min_{\boldsymbol{p}} \sum_{i=1}^{N} \frac{\pi_{1:T}^2(i)}{\boldsymbol{p}_i} = \frac{\left( \sum_{i=1}^{N} \sqrt{\pi_{1:T}^2(i)} \right)^2}{K} \\
&\leq \frac{T}{K} \left( \sum_{i=1}^{N} \pi_*(i) + \sum_{i=1}^{N} \sqrt{\frac{V_T(i)}{T}} \right)^2 \\
&= \frac{T}{K} \left( \Pi_*^2 + 2\Pi_* \sum_{i=1}^{N} \sqrt{\frac{V_T(i)}{T}} + \Big( \sum_{i=1}^{N} \sqrt{\frac{V_t(i)}{T}} \Big)^2 \right),
\end{aligned}
\tag{38}
$$

where we use Lemma 2.2 in the second line, and Equation (36) in the third line.

Then, we bound the second term in Equation (35):

$$
\begin{aligned}
\Pi_*^2 = \sum_{i=1}^{N} \pi_*^2(i) &\leq \left( \frac{1}{T} \sum_{t=1}^{T} \sum_{i=1}^{N} \pi_t(i) \right)^2 \leq \frac{1}{T} \sum_{t=1}^{T} (\sum_{i=1}^{N} \pi_t(i))^2 \\
&= \frac{K}{T} \sum_{t=1}^{T} \min_{\boldsymbol{p}} \sum_{i=1}^{N} \frac{\pi_t^2(i)}{\boldsymbol{p}_i},
\end{aligned}
\tag{39}
$$

where the first inequality uses the average assumption, the third inequality uses Jensen's inequality, and the last inequality uses Equation (37).

Overall, we combine the results in Equation (38) and Equation (39), and conclude the proof:

$$\min_{\boldsymbol{p}} \sum_{t=1}^{T} \ell_t(\boldsymbol{p}) - \sum_{t=1}^{T} \min_{\boldsymbol{p}} \ell_t(\boldsymbol{p}) \leq \frac{T}{K} \left( \sum_{i=1}^{N} \sqrt{\frac{V_T(i)}{T}} \right) \left( 2\Pi_* + \sum_{i=1}^{N} \sqrt{\frac{V_t(i)}{T}} \right).$$

### D.2 Regret of Full Information

In this section, we prove the preliminary theorem for Theorem 5.2. We first investigate the upper bound of $\text{Regret}_S(T)$ in an ideal scenario called *full feedback*, where the server preserves feedback information of all clients, i.e., $\{\pi_\tau(i)\}_{\tau=1}^{t-1}, \forall i \in [N], t \in [T]$. Technically, we prove this bound follows the framework of FTRL in online convex optimization, w.r.t sampling probability. Then, we extend it into practical settings in the next section and derive the expected regret bound.

**Theorem D.1** (Static regret with full information). *Under Assumptions 5.1, sampling a batch of clients with an expected size of $K$, and setting $\gamma = G^2$, the FTRL scheme in Equation (10) yields the following regret,*

$$\sum_{t=1}^{T} \ell_t(\boldsymbol{p}^t) - \min_{\boldsymbol{p}} \sum_{t=1}^{T} \ell_t(\boldsymbol{p}) \leq \left( \frac{22NG}{\bar{z}} + \frac{2\sqrt{6}NG}{K} \right) \sum_{i=1}^{N} \sqrt{\pi_{1:T}^2(i)} + \frac{22NG^2}{\bar{z}}, \tag{40}$$

*where we note the cumulative feedback $\sqrt{\pi_{1:T}^2(i)} \leq \mathcal{O}(\sqrt{T})$ following Assumption 5.1.*

*Proof.* We consider a restricted probability space $\Delta = \{\boldsymbol{p} \in \mathbb{R}^N | \boldsymbol{p}_i \geq p_{\min}, \sum_{i=1}^{N} \boldsymbol{p}_i = K, \forall i \in [N]\}$ where $p_{\min} \leq K/N$. Then, we decompose the regret,

$$\text{Regret}_S(T) = \underbrace{\sum_{t=1}^{T} \ell_t(\boldsymbol{p}^t) - \min_{p \in \Delta} \sum_{t=1}^{T} \ell_t(\boldsymbol{p})}_{(A)} + \underbrace{\min_{p \in \Delta} \sum_{t=1}^{T} \ell_t(\boldsymbol{p}) - \min_{\boldsymbol{p}} \sum_{t=1}^{T} \ell_t(\boldsymbol{p})}_{(B)}. \tag{41}$$

We separately bound the above terms in this section. The bound of (A) is related to the stability of the online decision sequence by playing FTRL, which is given in Lemma D.1. The minimal results of directing calculation bound term (B).

**Bounding (A)**. Without loss of generality, we introduce the stability of the online decision sequence from FTRL to cost function $\ell$ as shown in the following lemma(Kalai & Vempala, 2005) (similar proof can also be found in (Hazan, 2012; Shalev-Shwartz et al., 2012)).

**Lemma D.1.** *Let $\mathcal{K}$ be a convex set and $\mathcal{R} : \mathcal{K} \mapsto \mathbb{R}$ be a regularizer. Given a sequence of functions $\{\ell_t\}_{t \in [T]}$ defined over $\mathcal{K}$, then setting $\boldsymbol{p}^t = \arg\min_{\boldsymbol{p} \in \mathbb{R}^N} \sum_{\tau=1}^{t-1} \ell_\tau(\boldsymbol{p}) + \mathcal{R}(\boldsymbol{p})$ ensures,*

$$\sum_{t=1}^{T} \ell_t(\boldsymbol{p}^t) - \sum_{t=1}^{T} \ell_t(\boldsymbol{p}) \leq \sum_{t=1}^{T}(\ell_t(\boldsymbol{p}^t) - \ell_t(\boldsymbol{p}^{t+1})) + (\mathcal{R}(\boldsymbol{p}) - \mathcal{R}(\boldsymbol{p}^1)), \forall \boldsymbol{p} \in \mathcal{K}.$$

We note that $\mathcal{R}(\boldsymbol{p}) = \sum_{i=1}^{N} \gamma/\boldsymbol{p}_i$ in our work. Furthermore, $\mathcal{R}(\boldsymbol{p})$ is non-negative and bounded by $N\gamma/p_{\min}$ with $p \in \Delta$. Thus, the above lemma incurs,

$$\sum_{t=1}^{T} \ell_t(\boldsymbol{p}^t) - \sum_{t=1}^{T} \ell_t(\boldsymbol{p}) \leq \underbrace{\sum_{t=1}^{T}(\ell_t(\boldsymbol{p}^t) - \ell_t(\boldsymbol{p}^{t+1}))}_{\text{Bounded Below}} + \frac{N\gamma}{p_{\min}}. \tag{42}$$

To simply the following proof, we assume that $0 < \pi_1(t) \leq \pi_2(t) \leq \cdots \leq \pi_N(t), t \in [T]$ to satisfies Lemma B.8 without the loss of generality. The stability relies on the evolution of cumulative feedback $\pi_{1:t}^2(i)$ and hence relies on the index in solution $l_1, l_2$ according to Lemma 2.2. Following the Lemma B.8, we have

$$\boldsymbol{p}_i^t = \begin{cases} 1, & \text{if } i \geq l_2^t, \\ z_t \frac{\sqrt{\pi_{1:t-1}^2(i)+\gamma}}{c_t}, & \text{if } i \in (l_1^t, l_2^t), \\ p_{\min}, & \text{if } i \leq l_1^t, \end{cases} \tag{43}$$

where $z_t = K - N + l_2^t - l_1^t \cdot p_{\min} \leq K$ and $c_t = \sum_{i \in (l_1^t, l_2^t)} \sqrt{\pi_{1:t}^2(i) + \gamma} \leq \sum_{i=1}^{N} \sqrt{\pi_{1:t}^2(i) + \gamma}$ is the normalization factor . Then, we investigate the first term in the above inequality,

$$\sum_{t=1}^{T}(\ell_t(\boldsymbol{p}^t) - \ell_t(\boldsymbol{p}^{t+1})) \leq \sum_{t=1}^{T} \sum_{i=1}^{N} \pi_t^2(i) \cdot \left( \frac{1}{\boldsymbol{p}_i^t} - \frac{1}{\boldsymbol{p}_i^{t+1}} \right).$$

**Remark**. According to the above inequality, we note that the stability of online convex optimization is highly related to the changing probability. We can have a trivial upper bound $\sum_{t=1}^{T}(\ell_t(\boldsymbol{p}^t) - \ell_t(\boldsymbol{p}^{t+1})) \leq \sum_{t=1}^{T} \sum_{i=1}^{N} \pi_t^2(i) \cdot (1/p_{\min} - 1)$, which indicates that the stability is restricted by $p_{\min}$. Solving the sampling probability requires sorting cumulative feedbacks $\pi_{1:t}^2(i)$, the combinations of client-index and $\boldsymbol{p}_i^t$ are dynamic. Hence, directly bounding the above equation generally can be difficult. To obtain a tighter bound for FTRL, we investigate the possibility.

**Lemma D.2.** *Assuming that $\boldsymbol{p}_i^t < \boldsymbol{p}_i^{t+1}$, for all $i \in [N], t \in [T-1]$, the upper bound of $\left( \frac{1}{\boldsymbol{p}_i^t} - \frac{1}{\boldsymbol{p}_i^{t+1}} \right)$ is given by:*

$$0 \leq \left( \frac{1}{\boldsymbol{p}_i^t} - \frac{1}{\boldsymbol{p}_i^{t+1}} \right) \leq \frac{1}{\min(z_t, z_{t+1})} \left( \frac{c_t}{\sqrt{\pi_{1:t-1}^2(i) + \gamma}} - \frac{c_{t+1}}{\sqrt{\pi_{1:t}^2(i) + \gamma}} \right). \tag{44}$$

*Proof.* For all $t \in [T]$, we have cumulative feedbacks $\pi_{1:t-1}(i)$, $i \in [N]$ on the server. The server is able to compute results Equation (11). As we are interested in the upper bound, we assume $\boldsymbol{p}_i^t < \boldsymbol{p}_i^{t+1}$ and discuss the cases below:

- **Case 1**: letting $(\boldsymbol{p}_i^t, \boldsymbol{p}_i^{t+1}) = (p_{\min}, z_{t+1} \frac{\sqrt{\pi_{1:t}^2(i)+\gamma}}{c_{t+1}})$, we have

$$
\frac{1}{\boldsymbol{p}_i^t} - \frac{1}{\boldsymbol{p}_i^{t+1}} = \frac{1}{p_{\min}} - \frac{c_{t+1}}{z_{t+1}\sqrt{\pi_{1:t}^2(i)+\gamma}}
$$
$$
\leq \frac{c_t}{z_t\sqrt{\pi_{1:t-1}^2(i)+\gamma}} - \frac{c_{t+1}}{z_{t+1}\sqrt{\pi_{1:t}^2(i)+\gamma}},
$$
$$
\leq \frac{1}{\min(z_t, z_{t+1})} \left( \frac{c_t}{\sqrt{\pi_{1:t-1}^2(i)+\gamma}} - \frac{c_{t+1}}{\sqrt{\pi_{1:t}^2(i)+\gamma}} \right),
$$

where the second inequality uses Equation (24) indicating $p_{\min} \geq z_t \frac{\sqrt{\pi_{1:t-1}^2(i)+\gamma}}{c_t}$.

- **Case 2**: letting $(\boldsymbol{p}_i^t, \boldsymbol{p}_i^{t+1}) = (z_t \frac{\sqrt{\pi_{1:t-1}^2(i)+\gamma}}{c_t}, z_{t+1} \frac{\sqrt{\pi_{1:t}^2(i)+\gamma}}{c_{t+1}})$, Equation (44) naturally holds.

- **Case 3**: letting $(\boldsymbol{p}_i^t, \boldsymbol{p}_i^{t+1}) = (z_t \frac{\sqrt{\pi_{1:t-1}^2(i)+\gamma}}{c_t}, 1)$, we can know that $1 \leq z_{t+1} \frac{\sqrt{\pi_{1:t}^2(i)+\gamma}}{c_{t+1}}$ by Equation (24) and prove the conclusion analogous to case 1.

- **Case 4**: analogous to the case 1 and 3, letting $(\boldsymbol{p}_i^t, \boldsymbol{p}_i^{t+1}) = (p_{\min}, 1)$, Equation (44) naturally holds.

Summarizing all cases to conclude the proof. □

Using Lemma D.2, we are ready to bound the stability of the online decision sequence:

$$
\sum_{t=1}^{T}(\ell_t(\boldsymbol{p}^t) - \ell_t(\boldsymbol{p}^{t+1})) = \sum_{t=1}^{T}\sum_{i=1}^{N} \pi_t^2(i) \cdot \left( \frac{c_t}{z_t\sqrt{\pi_{1:t-1}^2(i)+\gamma}} - \frac{c_{t+1}}{z_{t+1}\sqrt{\pi_{1:t}^2(i)+\gamma}} \right)
$$
$$
\leq \sum_{t=1}^{T}\sum_{i=1}^{N} \frac{\pi_t^2(i) \cdot c_t}{\min(z_t, z_{t+1})} \cdot \left( \frac{1}{\sqrt{\pi_{1:t-1}^2(i)+\gamma}} - \frac{1}{\sqrt{\pi_{1:t}^2(i)+\gamma}} \right) \qquad \triangleright c_t \leq c_{t+1}
$$
$$
\leq \sum_{t=1}^{T}\sum_{i=1}^{N} \frac{\pi_t^2(i) \cdot \tilde{c}_t}{\min(z_t, z_{t+1})\sqrt{\pi_{1:t}^2(i)+\gamma}} \cdot \left( \sqrt{1 + \frac{\pi_t^2(i)}{\pi_{1:t-1}^2(i)+\gamma}} - 1 \right)
$$
$$
\leq \frac{\tilde{c}_T}{2} \sum_{t=1}^{T}\sum_{i=1}^{N} \frac{1}{\min(z_t, z_{t+1})} \frac{\pi_t(i)^4}{\sqrt{\pi_{1:t}^2(i)+\gamma} \cdot (\pi_{1:t-1}^2(i)+\gamma)}, \qquad \triangleright \sqrt{1+x}-1 \leq \frac{x}{2}
$$

where the third line uses definition $c_t \leq \tilde{c}_t = \sum_{i=1}^{N}\sqrt{\pi_{1:t}^2(i)+\gamma}$.

Letting $\gamma = G^2$, we have that $\pi_{1:t}^2(i) \leq \pi_{1:t-1}^2(i) + \gamma$ and $\sqrt{\pi_{1:t}^2(i)} \leq \sqrt{\pi_{1:t}^2(i)+\gamma}$. We define $\bar{z} = \min\{z_t\}_{t=1}^{T}$ and conclude the bound,

$$
\sum_{t=1}^{T}(\ell_t(\boldsymbol{p}^t) - \ell_t(\boldsymbol{p}^{t+1})) \leq \frac{\tilde{c}_T}{2} \sum_{t=1}^{T}\sum_{i=1}^{N} \frac{\pi_t(i)^4}{(\pi_{1:t}^2(i))^{\frac{3}{2}}}
$$
$$
= G \cdot \frac{\tilde{c}_T}{2\bar{z}} \sum_{i=1}^{N}\sum_{t=1}^{T} \frac{(\pi_t(i)/G)^4}{((\pi_{1:t}(1)/G)^2)^{\frac{3}{2}}} \qquad \triangleright \text{Lemma B.1}
$$
$$
\leq \frac{22NG}{\bar{z}} \sum_{i=1}^{N}\sqrt{\pi_{1:T}^2(i)+G^2} \qquad \triangleright \text{Definition of } \tilde{c}_T
$$
$$
\leq \frac{22NG}{\bar{z}} \sum_{i=1}^{N}\left( \sqrt{\pi_{1:T}^2(i)} + G \right)
$$

$$\tag{45}$$

Finally, we can get the final bound of (A) by plugging Equation (45) into Equation (42) and summarizing as follows,

$$\sum_{t=1}^{T} \ell_t(\boldsymbol{p}^t) - \sum_{t=1}^{T} \ell_t(\boldsymbol{p}) \leq \frac{22NG}{\bar{z}} \sum_{i=1}^{N} \left( \sqrt{\pi_{1:T}^2(i)} + G \right) + \frac{NG^2}{p_{\min}}.$$

**Bounding (B)**. Using Corollaries B.1 and B.2, we bound the term (B) as follows,

$$
\begin{aligned}
&\min_{p \in \Delta} \sum_{t=1}^{T} \ell_t(\boldsymbol{p}) - \min_{\boldsymbol{p}} \sum_{t=1}^{T} \ell_t(\boldsymbol{p}) \\
&\leq \frac{K(\sum_{i=1}^{N} \sqrt{\pi_{1:T}^2(i)})^2}{(K - Np_{\min})^2} - \frac{(\sum_{i=1}^{N} \sqrt{\pi_{1:T}^2(i)})^2}{K} \\
&\leq \left( \frac{K}{(K - Np_{\min})^2} - \frac{1}{K} \right) \cdot \left( \sum_{i=1}^{N} \sqrt{\pi_{1:T}^2(i)} \right)^2 \\
&\leq \frac{6Np_{\min}}{K^2} \cdot \left( \sum_{i=1}^{N} \sqrt{\pi_{1:T}^2(i)} \right)^2
\end{aligned}
\tag{46}
$$

In the last line, we use the fact that $\frac{1}{(1-x)^2} - 1 \leq 6x$ for $x \in [0, 1/2]$. Hence, we scale the coefficient

$$\frac{K}{(K - Np_{\min})^2} - \frac{1}{K} = \frac{1}{K} \left[ \frac{1}{(1 - Np_{\min}/K)^2} - 1 \right] \leq \frac{6Np_{\min}}{K^2},$$

where we let $p_{\min} \leq K/(2N)$.

**Summary**. Setting $\gamma = G^2$, and combining the bound in Equation (42) and Equation (46), we have,

$$
\begin{aligned}
\text{Regret}_{\text{S}}(T) &= \sum_{t=1}^{T} \ell_t(\boldsymbol{p}^t) - \min_{\boldsymbol{p}} \sum_{t=1}^{T} \ell_t(\boldsymbol{p}) \\
&\leq \frac{22NG}{\bar{z}} \sum_{i=1}^{N} \left( \sqrt{\pi_{1:T}^2(i)} + G \right) + \frac{NG^2}{p_{\min}} + \frac{6Np_{\min}}{K^2} \cdot \left( \sum_{i=1}^{N} \sqrt{\pi_{1:T}^2(i)} \right)^2.
\end{aligned}
\tag{47}
$$

The $p_{\min}$ is only relevant for the theoretical analysis. Hence, the choice of it is arbitrary, and we can set it to $p_{\min} = \min \left\{ K/(2N), GK/(\sqrt{6} \sum_{i=1}^{N} \sqrt{\pi_{1:T}^2(i)}) \right\}$ which turns the upper bound to the minimal value. Hence, we yield the final bound of FTRL in the end,

$$\sum_{t=1}^{T} \ell_t(\boldsymbol{p}^t) - \min_{\boldsymbol{p}} \sum_{t=1}^{T} \ell_t(\boldsymbol{p}) \leq \left( \frac{22NG}{\bar{z}} + \frac{2\sqrt{6}NG}{K} \right) \sum_{i=1}^{N} \sqrt{\pi_{1:T}^2(i)} + \frac{22NG^2}{\bar{z}}.
\tag{48}$$

$\square$

### D.3 Expected Regret of Partial Feedback: Proof of Theorem 5.2

In this section, we extend the full feedback solution to the partial feedback scenario, where the server only has access to the feedback information from the past sampled clients. Technically, we use a mixing strategy to transfer the deterministic regret into expected regret. Then, we bound the expected regret by playing the FTRL framework in the expectation space. Finally, we can minimize the expected regret bound by setting mixing parameter $\theta$.

Using the property of unbiasedness, we have

$$
\min_{\boldsymbol{p}} \mathbb{E}[\sum_{t=1}^{T} \ell_t(\tilde{\boldsymbol{p}}^t) - \sum_{t=1}^{T} \ell_t(\boldsymbol{p})]
$$
$$
= \min_{\boldsymbol{p}} \mathbb{E}[\sum_{t=1}^{T} \tilde{\ell}_t(\tilde{\boldsymbol{p}}^t) - \sum_{t=1}^{T} \tilde{\ell}_t(\boldsymbol{p})] \tag{49}
$$
$$
= \underbrace{\mathbb{E}\Big[ \sum_{t=1}^{T} \tilde{\ell}_t(\tilde{\boldsymbol{p}}^t) - \sum_{t=1}^{T} \tilde{\ell}_t(\boldsymbol{p}^t) \Big]}_{(A)} + \underbrace{\min_{\boldsymbol{p}} \mathbb{E}\Big[ \sum_{t=1}^{T} \tilde{\ell}_t(\boldsymbol{p}^t) - \sum_{t=1}^{T} \ell_t(\boldsymbol{p}) \Big]}_{(B)}.
$$

**Bounding (A)**. We recall that $\tilde{\boldsymbol{p}}_i^t \geq \frac{\theta K}{N}$ for all $t \in [T], i \in [N]$ due to the mixing. Therefore, $\boldsymbol{p}_i^t \geq K/N$ implies $\tilde{\boldsymbol{p}}_i^t \geq K/N$. Thus, we have

$$
\frac{1}{\tilde{\boldsymbol{p}}_i^t} - \frac{1}{\boldsymbol{p}_i^t} = \theta \cdot \frac{\boldsymbol{p}_i^t - \frac{K}{N}}{\tilde{\boldsymbol{p}}_i^t \boldsymbol{p}_i^t} \leq \theta \cdot \frac{\boldsymbol{p}_i^t}{\tilde{\boldsymbol{p}}_i^t \boldsymbol{p}_i^t} = \frac{\theta}{\tilde{\boldsymbol{p}}_i^t} \leq \theta \cdot \frac{N}{K}.
$$

Moreover, if $\boldsymbol{p}_i^t \leq K/N$, the above inequality still holds. We extend the (A) as follows,

$$
(A) := \mathbb{E}\Big[ \sum_{t=1}^{T} \tilde{\ell}_t(\tilde{\boldsymbol{p}}^t) - \sum_{t=1}^{T} \tilde{\ell}_t(\boldsymbol{p}^t) \Big]
$$
$$
= \mathbb{E}\Big[ \sum_{t=1}^{T} \sum_{i=1}^{N} \tilde{\pi}_t^2(i) \Big( \frac{1}{\tilde{\boldsymbol{p}}_i^t} - \frac{1}{\boldsymbol{p}_i^t} \Big) \Big]
$$
$$
\leq \theta \cdot \frac{N}{K} \cdot \mathbb{E}\Big[ \sum_{t=1}^{T} \sum_{i=1}^{N} \tilde{\pi}_t^2(i) \Big]
$$
$$
\leq \frac{\theta G^2 N^2}{K} T,
$$

where we use $\mathbb{E}[\tilde{\pi}_t^2(i)] = \pi_t^2(i) \leq G^2$.

**Bounding (B)**. We note that $\boldsymbol{p}^t$ is the decision sequence playing FTRL with the mixed cost functions. Thus, we combine the mixing bound of feedback (i.e., $\tilde{\pi}_t^2(i) \leq \frac{G^2 N}{\theta K}$) and Theorem D.1. Replacing $G^2$ with $G^2 \frac{N}{\theta K}$, we get

$$
\sum_{t=1}^{T} \tilde{\ell}_t(\boldsymbol{p}^t) - \min_{\boldsymbol{p}} \sum_{t=1}^{T} \tilde{\ell}_t(\boldsymbol{p}) \leq \left( \frac{22 N^{\frac{3}{2}} G}{\bar{z}\sqrt{\theta K}} + \frac{2\sqrt{6} N^{\frac{3}{2}} G}{\sqrt{\theta K^3}} \right) \mathbb{E}\left[ \sum_{i=1}^{N} \sqrt{\tilde{\pi}_{1:T}^2(i)} \right] + \frac{22 G^2 N^2}{\bar{z}\theta K}. \tag{50}
$$

**Summary**. Using Jensen's inequality, we have $\mathbb{E}\big[ \sum_{i=1}^{N} \sqrt{\tilde{\pi}_{1:T}^2(i)} \big] \leq \sum_{i=1}^{N} \sqrt{\mathbb{E}[\tilde{\pi}_{1:T}^2(i)]} = \sum_{i=1}^{N} \sqrt{\pi_{1:T}^2(i)}$. Finally, we can get the upper bound of the regret in partial-bandit feedback,

$$
N^2 \cdot \min_{\boldsymbol{p}} \mathbb{E}[\sum_{t=1}^{T} \ell_t(\tilde{\boldsymbol{p}}^t) - \sum_{t=1}^{T} \ell_t(\boldsymbol{p})] \leq \frac{\theta G^2}{K} T + \left( \frac{22 G}{\bar{z}\sqrt{\theta N K}} + \frac{2\sqrt{6} G}{\sqrt{\theta N K^3}} \right) \mathbb{E}\left[ \sum_{i=1}^{N} \sqrt{\tilde{\pi}_{1:T}^2(i)} \right] + \frac{22 G^2}{\bar{z}\theta K}
$$
$$
\leq \frac{\theta G^2}{K} T + \left( \frac{22 N^{\frac{1}{2}} G^2}{\bar{z}\sqrt{\theta K}} + \frac{2\sqrt{6} N^{\frac{1}{2}} G^2}{\sqrt{\theta K^3}} \right) \sqrt{T} + \frac{22 G^2}{\bar{z}\theta K}, \tag{51}
$$

where the last line uses the bound $\sum_{i=1}^{N} \sqrt{\pi_{1:T}^2(i)} \leq N G \sqrt{T}$. Now, we can optimize the upper bound of regret in terms of $\theta$. Notably, $\theta$ is independent on $T$ and we set $\theta = (\frac{N}{TK})^{\frac{1}{3}}$ to get the minimized bound. Additionally,

we are pursuing an expected regret, which is $\mathrm{Regret}_{(S)}(T)$ in the original definition in Equation (9). Using the unbiasedness of the mixed estimation and modified costs, we can obtain the final bound:

$$
\begin{aligned}
N^2 \cdot \mathbb{E}[\mathrm{Regret}_{(S)}(T)] &= \mathbb{E}[\sum_{t=1}^{T} \ell_t(\tilde{\boldsymbol{p}}^t) - \min_{\boldsymbol{p}} \sum_{t=1}^{T} \ell_t(\boldsymbol{p})] \\
&= \mathbb{E}[\sum_{t=1}^{T} \ell_t(\tilde{\boldsymbol{p}}^t) - \min_{\boldsymbol{p}} \sum_{t=1}^{T} \tilde{\ell}_t(\boldsymbol{p})] + \mathbb{E}[\min_{\boldsymbol{p}} \sum_{t=1}^{T} \tilde{\ell}_t(\boldsymbol{p}) - \min_{\boldsymbol{p}} \sum_{t=1}^{T} \ell_t(\boldsymbol{p})] \\
&\leq \mathcal{O}\big(N^{\frac{1}{3}} T^{\frac{2}{3}} / K^{\frac{4}{3}}\big) + \mathbb{E}[\min_{\boldsymbol{p}} \sum_{t=1}^{T} \tilde{\ell}_t(\boldsymbol{p}) - \min_{\boldsymbol{p}} \sum_{t=1}^{T} \ell_t(\boldsymbol{p})] \\
&\leq \tilde{\mathcal{O}}\big(N^{\frac{1}{3}} T^{\frac{2}{3}} / K^{\frac{4}{3}}\big),
\end{aligned}
$$

where the last inequality uses Lemma 5.1, and the conclusion in Theorem 8 (Borsos et al., 2018). It proves the second term induces an additional log term to the final bound.

**Remark.** Baseline works have additional averaging coefficient $\frac{1}{N^2}$ in their final bound. This is because they consider the weights $\boldsymbol{\lambda} = 1/N$ in stochastic optimization, while we include the $\lambda$ for clients' weights in federated optimization. To align with them, we omit the coefficient of $N^2$ and report the final bound for $\mathbb{E}[\mathrm{Regret}_{(S)}(T)]$, as $N^2$ can be absorbed by excluding the $\boldsymbol{\lambda}$ from client feedback function $\pi(\cdot)$.

# E   Further Discussions

## E.1   A Sketch of Proof for FL with Client Stragglers

We note the possibility that some clients are unavailable to participants due to local failure or being busy in each round. To extend our analysis to the case, we assume there is a known distribution of client availability $\mathcal{A}$ such that a subset $\mathcal{A}^t \sim \mathcal{A}$ of clients are available at the $t$-th communication round. Let $\boldsymbol{q}_i = \mathrm{Prob}(i \in \mathcal{A}^t)$ denote the probability that client $i$ is available at round $t$. Based on the setting, we update the definition of estimation $\boldsymbol{d}^t$:

$$
\boldsymbol{d}^t := \sum_{i \in S^t \subseteq \mathcal{A}^t} \frac{\boldsymbol{\lambda}_i \boldsymbol{g}_i^t}{\boldsymbol{q}_i \boldsymbol{p}_i^t},
$$

where $S^t \subseteq \mathcal{A}^t$ indicates that we can only sample from available set. Then, we apply the estimation to variance and obtain the following target:

$$
\mathrm{Regret}(T) = \sum_{t=1}^{T} \sum_{i=1}^{N} \frac{\pi_t^2(i)}{\boldsymbol{q}_i \boldsymbol{p}_i} - \sum_{t=1}^{T} \min_{\boldsymbol{p}} \sum_{i=1}^{N} \frac{\pi_t^2(i)}{\boldsymbol{q}_i \boldsymbol{p}_i}.
$$

Analogous to our analysis in Appendix D, we could obtain a similar bound of the above regret that considers the availability.

## E.2   Hard Capping for ISP

We know that ISP might have more or less than $K$ clients selected, which implies the sampling results $S^t$ can be unexpectedly large. Concretely, this section studies a setting in which RSP with hard budget $K_h$ is compared a soft budget $K_s$ to ensure that $\mathrm{Prob}(|S^t| > K_h) \leq \epsilon$. We intend to analyze the constraints on sampling probability $\boldsymbol{p}$ w.r.t $K_s, K_h$ and $\epsilon$. This objective can be formally described as:

$$
\mathrm{Prob}(|S^t| > K_h) = 1 - \mathrm{Prob}(|S^t| \leq K_h) = 1 - \sum_{k=0}^{K_h} \sum_{C \in C_N^k} \left( \prod_{i \in C} p_i \prod_{j \notin C} (1 - p_j) \right) \leq \epsilon, \tag{52}
$$

where $\sum_{i=1}^{N} \boldsymbol{p}_i = K_s$, $k$ is the possible size of $S^t$ and $C_N^k$ is the combinations of all possible selected clients with $|S^t| = k$. If we consider a simple case where $\boldsymbol{p}_i = K_s/N, \forall i \in [N]$, $|S^t|$ follows standard Binomial

distribution $B(N, K_s/N)$. And, the above inequality can be solved by checking the cumulative distribution function (CDF).

For general cases with different $\boldsymbol{p}_i, \forall i \in [N]$, we can use Normal approximation to solve the inequality with large $N$. Using Hoeffding's inequality to estimate the probability of large deviations:

$$\text{Prob}(|S^t| - K_s \geq \tau) \leq e^{-2\tau^2/\sum_{i=1}^N \boldsymbol{p}_i(1-\boldsymbol{p}_i)}.$$

Letting $K_h = K_s + \tau$, we know

$$\text{Prob}(|S^t| \geq K_h) \leq e^{-2\tau^2/\sum_{i=1}^N \boldsymbol{p}_i(1-\boldsymbol{p}_i)} \leq \epsilon.$$

Taking the logarithm and rearranging the second inequality, we get

$$\sum_{i=1}^N \boldsymbol{p}_i(1 - \boldsymbol{p}_i) \geq -2(K_h - K_s)^2/\ln \epsilon,$$

as a new constraint for ISP sampling probability to satisfy Equation (52). In other words, adding this constraint on Equation (22) may result in a satisfied sampling probability.

### E.3 Differences between biased client sampling methods

This section discusses the main differences between unbiased client sampling and biased client sampling methods. The proposed K-Vib sampler is an unbiased sampler for the first-order gradient of objective 1. Recent biased client sampling methods include Power-of-Choice (POC) (Cho et al., 2020b) and DivFL (Balakrishnan et al., 2022). Concretely, POC requires all clients to upload local empirical loss as prior knowledge and selects clients with the largest empirical loss. DivFL builds a submodular based on the latest gradient from clients and selects clients to approximate all client information. Therefore, these client sampling strategies build a biased gradient estimation that may deviate from a fixed global goal.

FL with biased client sampling methods, such as POC and DivFL, can be considered dynamic re-weighting algorithms adjusting $\boldsymbol{p}_i$. Analogous to the Equation (1), the basic objective of FL with biased client sampling methods can be defined as follows (Li et al., 2020b; Balakrishnan et al., 2022; Cho et al., 2020b):

$$\min_{\boldsymbol{x} \in \mathcal{X}} f(\boldsymbol{x}) := \sum_{i=1}^N \boldsymbol{p}_i f_i(\boldsymbol{x}) := \sum_{i=1}^N \boldsymbol{p}_i \mathbb{E}_{\xi_i \sim \mathcal{D}_i}[F_i(\boldsymbol{x}, \xi_i)], \tag{53}$$

where $\boldsymbol{p}$ is the probability simplex, and $\boldsymbol{p}_i$ is the probability of client $i$ being sampled. The gradient estimation is defined as $\boldsymbol{g}^t = \frac{1}{K} \sum_{i \in S^t} \boldsymbol{g}_i$ accordingly. The targets of biased FL client sampling are determined by the sampling probability $\boldsymbol{p}$ as a replacement of $\boldsymbol{\lambda}$ in the original FedAvg objective 1. Typically, the value of $\boldsymbol{p}$ is usually dynamic and implicit.

### E.4 Theoretical Comparison with OSMD

The K-Vib sampler proposed in this paper is orthogonal with the recent work OSMD sampler Zhao et al. (2021b)[2] in theoretical contribution. We justify our points below:

a) According to Equations (6) and (7) in OSMD, it proposes an online mirror descent procedure that optimizes the additional estimates to replace the mixing strategy in Vrb Borsos et al. (2018). The approach can be also utilized as an alternative method in Equation (12).

b) The improvement of the K-Vib sampler is obtained from the modification of the sampling procedure. In contrast, the OSMD still follows the conventional RSP, as we discussed in Lemma 2.1. Hence, our theoretical findings of applying the ISP in adaptive client sampling can be transferred to OSMD as well.

In short, the theoretical improvement of our work is independent of the OSMD sampler. And, our insights about utilizing the ISP can be used to improve the OSMD sampler. Meanwhile, the OSMD also suggests future work for the K-Vib sampler in optimizing the additional estimates procedure instead of mixing.

---

[2]we refer to the latest version `https://arxiv.org/pdf/2112.14332.pdf`

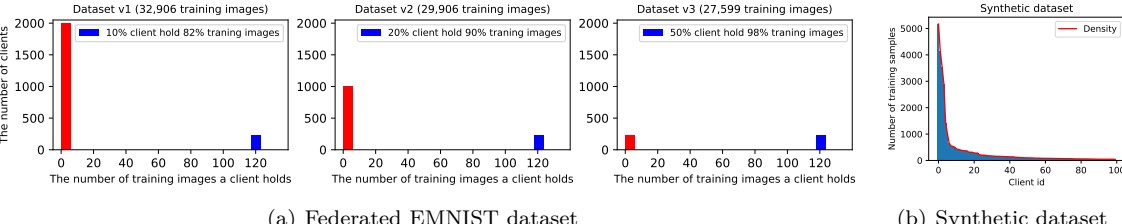

(a) Federated EMNIST dataset          (b) Synthetic dataset

Figure 6: Distribution of used datasets across clients.

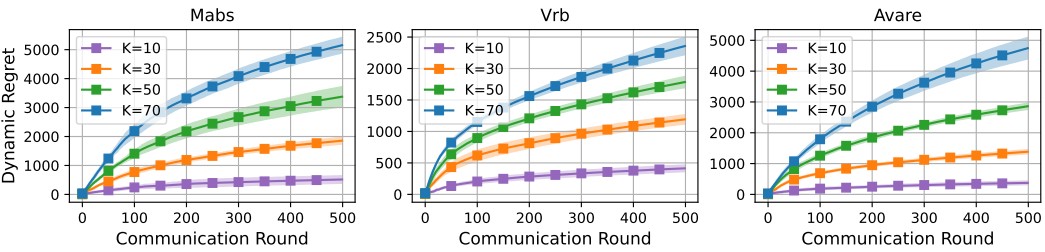

Figure 7: Regret of baseline algorithms with different $K$

# F Further Experiments

The experiment implementations are supported by *FedLab* framework (Zeng et al., 2023b). We provide additional experimental details, experiments, and discussion in this section.

## F.1 Experiment Details

**Hyperparameters Setting**. All samplers have an implicit value $G$ related to the hyperparameters. We set $G = 0.01$ for the Synthetic dataset task and $G = 0.1$ for FEMNIST tasks. We set $\eta = 0.4$ (stability hyperparameter) for Mabs (Salehi et al., 2017) as suggested by the original paper. Vrb Borsos et al. (2018) also utilize mixing strategy $\theta = (N/T)^{\frac{1}{3}}$ and regularization $\gamma = G^2 * N/\theta$. For the case that $N > T$ in FEMNIST tasks, we set $\theta = 0.3$ following the official source code[3]. For Avare El Hanchi & Stephens (2020), we set $p_{\min} = \frac{1}{5N}$, $C = \frac{1}{\frac{1}{N} - p_{\min}}$ and $\delta = 1$ for constant-stepsize as suggested in Appendix D of original paper. For the K-Vib sampler, we set $\theta = (\frac{N}{TK})^{1/3}$ and $\gamma = G^2 \frac{N}{K\theta}$. We also fix $\gamma$ and $\theta = 0.3$ for our sensitivity study in Figure 3(d).

**Data distribution across clients.** We present the data distribution across clients in Figure 6. The data samples are extremely unbalanced and long-tailed, matching the real-world applications. It also shows the significance of variance reduction techniques.

## F.2 Additional Experiments

**Baselines with budget $K$.** Our theoretical results in Theorem 5.2 and empirical results in Figure 3(a) reveal a key improvement of our work, that is, the linear speed up in online convex optimization. In contrast, we provide additional experiments with the different budget $K$ in Figure 7. Baseline methods do not preserve the improvement property respecting large budget $K$ in adaptive client sampling for variance reduction. Moreover, with the increasing communication budget $K$, the optimal sampling value is decreasing. As a result, the regret of baselines increases in Figure 7, indicating the discrepancy to the optimal is enlarged.

**Total variation distance over communication rounds.** In Figure 9, we provide the TV between probabilities of K-Vib and uniform probabilities. It shows that the trends of TV distances are dependent on data distribution across clients. The TV distances change more with a large degree of heterogeneity across

---

[3]https://github.com/zalanborsos/online-variance-reduction

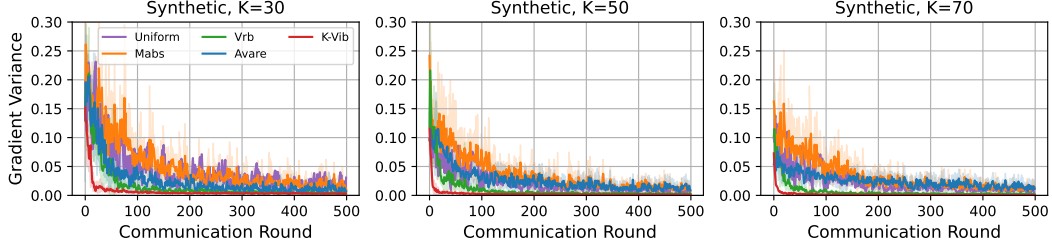

Figure 8: Gradient variance with different $K$

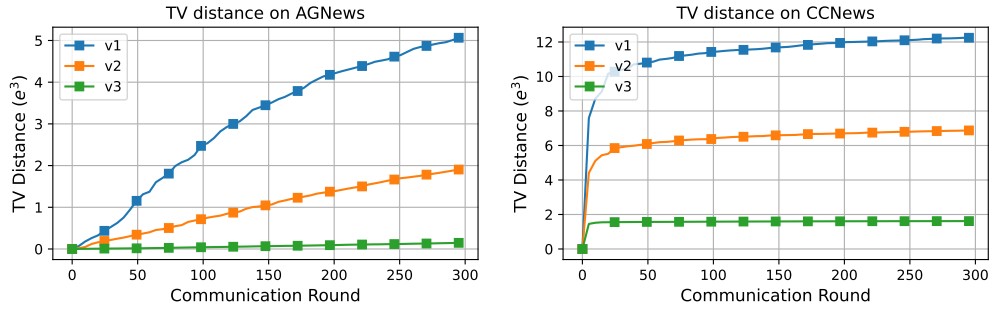

Figure 9: Evolution of TV distance between K-Vib probabilities and Uniform probabilities.

clients. This indicates that K-Vib can accurately be adaptive to data distributions across clients. Moreover, we note that the trends of TV curves are different in AGNews and CCNews, as their loss types (classification loss and next-token-prediction loss) and model architectures (Bert and GPT) are different.

### F.3 Efficient Implementation

In experiments, we do not find a heavy computation time increase compared to baselines as our experiments only involve thousands of clients. To guarantee practical usage for large-scale systems, we present efficient implementation details of K-Vib.

We can maintain a sorted list of cumulative local weights $[\omega(1), \omega(2), \ldots, \omega(N)]$ such that $\omega(i) \leq \omega(j)$, $\forall i, j \in [N]$ in Algorithm 2. For each communication round, the server receives feedback values as a list $[\pi_t(j)], \forall j \in S^t$. Then, the server will traverse the feedback list. For each element in the list, the server conducts two main steps as below:

- Step 1: For each $j \in S^t$, server computes estimates $\tilde{\omega}(j) = \omega(j) + \pi_t^2(j)/p_j^t$. Then, the server uses *binary-search* to find the index $k$ such that $\omega(k) \leq \tilde{\omega}(j) < \omega(k+1)$ in the cumulative local weights.

- Step 2: Then, server update $\omega(j) = \tilde{\omega}(j)$ and move the position of $\omega(j)$ behind $\omega(k)$ to update the weights sequence.

This implementation implements a time complexity of $\mathcal{O}(T \cdot K \cdot \log N)$, where $T$ is the communication round, $K$ is the communication budget, and $N$ is the number of clients. For each communication round $t \in [T]$, the server updates $K$ times of the list with each time cost $\mathcal{O}(\log N)$ to conduct one binary search.

