# OpenReview forum: "Enhanced Federated Optimization: Adaptive Unbiased Client Sampling with Reduced Variance"
_TMLR — Accepted by TMLR_

### Review · Reviewer_Bvvp · 2024-10-03

**Summary Of Contributions:**

The article addresses a significant challenge in Federated Learning (FL), particularly the client selection process, which directly affects the convergence rate and efficiency of FL models. The authors propose a new client sampling method, K-Vib, based on an independent, adaptive sampling procedure that reduces the variance in client selection, leading to improved convergence. The theoretical contribution is supported by empirical results showing that K-Vib significantly improves the speed of training compared to existing baselines.

**Audience:**

Yes

**Broader Impact Concerns:**

See above

**Claims And Evidence:**

Yes

**Requested Changes:**

_Broaden Experimental Comparisons:_ Expanding the set of baseline algorithms in the experiments will provide a more comprehensive evaluation of K-Vib’s relative performance.

**Strengths And Weaknesses:**

Given that I am not an expert in this field, I highlight the following:

__Strengths:__
1) _Novel Client Sampling Approach:_ The proposal of K-Vib introduces a new adaptive unbiased client sampling framework that is conceptually innovative. The independent sampling procedure and variance reduction strategy represent a meaningful improvement over conventional random sampling methods.

2) _Empirical Validation:_ The empirical studies convincingly demonstrate that K-Vib outperforms baseline algorithms, doubling the speed of training in the experiments. This practical evidence strengthens the validity of the theoretical claims and shows the potential impact of K-Vib on real-world FL systems.

__Weaknesses:__

1) _Limited Benchmarking:_ While the experimental results highlight the superiority of K-Vib over certain baselines, the set of comparison methods is relatively narrow. It would be useful to see K-Vib benchmarked against a wider range of advanced FL techniques.

---

> ### Author Response · Authors · 2024-10-15
> **Thank you for your feedback!**
>
> Dear reviewer,
>
> Thank you very much for your valuable feedback and constructive comments. We greatly appreciate your positive remarks on the novelty of the K-Vib framework and its practical effectiveness in FL systems.
>
> Regarding your concern about **limited benchmarking**, we would like to clarify that we compared K-Vib with the most relevant online variance reduction methods in this work. Specifically, the online variance reduction problem addressed in Eq. (8) plays a fundamental role in accelerating stochastic optimization algorithms. Solutions such as Mabs, Vrb, and Avare can be naturally extended to data sampling [1], coordinate sampling [2], and federated client sampling, without incurring additional communication or computational costs. We believe these baselines are well-aligned with the scope of our research and reflect the state-of-the-art in this domain.
>
> That said, we are open to further enhancing the experimental section to address your insightful comments better. If there are specific FL algorithms or advanced techniques you believe would enrich the comparison, we would greatly appreciate your suggestions. Your feedback will help us ensure that our work offers even more comprehensive and impactful insights for the community.
>
> Thank you once again for your thoughtful review. We look forward to your response and will incorporate any necessary revisions accordingly.
>
> Best regards,
>
> Authors
>
>
>
> References
>
> [1] Angelos Katharopoulos and François Fleuret. Not all samples are created equal: Deep learning with importance sampling. In International conference on machine learning, pp. 2525–2534. PMLR, 2018.
>
> [2] Peter Richtárik and Martin Takáč. On optimal probabilities in stochastic coordinate descent methods. Optimization Letters, 10:1233–1243, 2016a.

---

### Review · Reviewer_pLCp · 2024-10-22

**Summary Of Contributions:**

This paper studies client sampling procedures for federated learning (FL) and introduces K-Vib, an adaptive unbiased client sampler, which utilizes an independent sampling procedure (ISP) instead of traditional random sampling (RSP). In FL, multiple devices (clients) collaborate to train a global model without sharing their local data. Instead of sending data to a central server, clients perform local computations and send model updates to the server, which aggregates them into a global model. Sampling procedures are needed to determine how the server acquires updates from the clients.

The key contribution is the reduction in variance during client sampling, which improves convergence rates in federated optimization. The authors argue that K-Vib achieves an improved regret bound compared to baselines, showing near-linear speedups in communication efficiency. Empirical studies validate K-Vib’s superior performance across various FL tasks, particularly in cross-device settings with heterogeneous client data distributions.

**Audience:**

Yes

**Broader Impact Concerns:**

I don't see additional concerns that need to be addressed.

**Claims And Evidence:**

Yes

**Requested Changes:**

Critical:
- In order to better understand the significance of the contribution, I hope to see additional discussion on the two theoretical baselines used for comparison and on the empirical improvements. A better quantification of the improvement (e.g., comparing ISP vs RSP) can be helpful.
- Regarding Section 3 on cases study on sampling procedures, I am not sure if I am convinced with these toy examples, since after all we should be concerned with cases where $N$ and $K$ are both large. The examples on small universes may not provide insights that are robust when scaled up. More discussion on this is useful.
- I would love to see more discussion on the hyperparameters, $\theta$ in particular.

Minor:
- Many proofs in the appendix are not easily readable. It may help to break up long chains of equations and provide an overview at the beginning of long proofs.
- End of Section 1, paragraph "Contributions": The budget $K$ has not been defined at this point.
- Legend of Figure 1(b) seems erroneous: Where is ISP? Also, caption to the Figure is confusing (Is "Global estimates on the X-Y plane" a run-away statement?).
- Theorem 5.2, proof: Should be "Proof sketch" or "Sketch of proof".

**Strengths And Weaknesses:**

Strengths:
- The paper studies client sampling for FL, which is an important aspect of modern machine learning.
- The paper presents both theoretical analysis and empirical evaluations, which well demonstrate the usefulness of the proposed sampling algorithm. The theoretical parts seem overall correct and sound; the experiments are comprehensive and cover both synthetic and real world image and text data sets, touching on even large scale language models and data sets.

Weaknesses:
- Regarding the claim of superiority of K-Vib, I think more evidence may be necessary. First, it is unclear to me that the two referenced baselines are fair comparison, as they are not explicit handling the budget dependency term based on my understanding; it is unclear if there is a true improvement. On the empirical front, it is also not clear to me if the improvement is significant (e.g., Figure 2, many parts of Figure 5).
- Many of the visualizations seem hard to parse in my opinion. For instance, Figure 1(a) may be converted to histograms (rather than scatter plots) for better readability; Figure 4(a) seems slightly convoluted - I don't think the bars provide useful information, maybe just stating the data distribution suffices. In addition, there are occasional typos/minor errors (e.g., citation styles) that require further polishing.

---

> ### Author Response · Authors · 2024-10-30
> **Thank you for your feedback!**
>
> We would like to express our gratitude to the reviewer for the insightful comments and suggestions. We appreciate the positive feedback regarding the importance of our study and the strength of our theoretical and empirical evaluations. Understanding the need to clarify the significance of our contributions and provide informative materials, we have refined our work through additional clarification experiments and extended discussions.
>
> Below, we address the critical and minor concerns raised, and all revisions are marked in blue.
>
> **Critical and Weaknesses:**
>
> 1. **Comparing ISP vs RSP:** We provide a new Example 3.1, where we carefully compare the optimal estimates' variance of ISP and RSP. The new example reveals the variance reduction trends with different budgets $K$, demonstrating that ISP is more efficient than RSP under all budgets. Additionally, this example highlights a key drawback of RSP, as it sometimes produces invalid (non-variance-reduced) estimates. We hope this revised example reinforces our contribution to designing sampling probabilities for ISP over RSP in federated optimization.
>
> 2. **Toy Examples in Section 3:**   We have scaled up Example 3.1 to 1,000 vectors, aligning with the client size used in our empirical experiments. It also provides the mean variances of ISP and RSP from small to large budget $K$. We hope this new example provides valid insights into scaled-up federated optimization.
>
>    Based on the results, we have carefully refined the corresponding discussion. In particular, we highlight that ISP is better as it handles the budget $K$ better than RSP. We believe this new discussion can further clarify the theoretical insights in Lemma 2.2.
>
> 3. **Discussion on Hyperparameters:** We carefully reorganized our writing and provided clear implications of the hyperparameters of K-Vib at the end of Section 5. Theoretically, $\gamma$ is inherited from the FTRL framework to guarantee the stability of the designed probability sequence, and $\theta$ is optimized for minimizing the expected regret bound. Intuitively, K-Vib explores the system information during an early training period roughly using near-uniform probability. Thus, both hyperparameters can be viewed as tuning the span of the exploration period.
>
>    In Figure 3 (b), additional experiments show the regret value with different $\theta$. It suggests the importance of choosing proper $\theta$ for minimizing regret. Moreover, the results also match our theory which suggests setting $\theta = (\frac{N}{T K})^{1/3} = (\frac{100}{500\times 10})^{1/3} \approx 0.27$.
>
> 4. **The visualizations seem hard to parse.**   Thanks for the insightful feedback. We have modified Figure 1 (a) into histograms. And, we move the data distribution illustration of Synthetic and FEMNIST datasets to the Appendix and provide detailed discussion and additional results as a replacement.
>
> **Minor:**
>
> 1. **Readability of Proofs:** We appreciate the suggestion to improve the readability of the proofs in the appendix. We have added summaries for each subsection, broken down longer equation chains, and included comments on key derivation steps.
> 2. **Definition of Budget:** Thanks for point out. We have defined $K$ in the text.
> 3. **Legend of Figure 1(b):** We have redrawn Figure 1(b) and corrected its caption.
> 4. **Theorem 5.2:** Based on your feedback, we have updated the title in Theorem 5.2 to "Sketch of proof" to align with standard terminology.
> 5. **Additional minor revisions:** We have thoroughly proofread the draft and made further minor corrections to ensure it meets a high academic standard.
>
> In conclusion, we believe these revisions will strengthen our paper and enhance clarity. Thank you once again for your valuable feedback, which has greatly contributed to improving our work. **Most importantly, we are willing to make any improvements if required.**

---

> ### Author Response · Authors · 2024-11-25
> **Inquiry Regarding Satisfaction with Revised Manuscript**
>
> Dear reviewer,
>
> Thank you once again for your valuable feedback on our manuscript, which greatly helped us improve the quality of our work. We have carefully addressed all your comments and suggestions in the revised version and included a detailed response outlining the changes made.
>
> We kindly ask whether the revisions meet your expectations and if you have any additional comments or suggestions for further improvement. Please let us know if we should address any remaining concerns to ensure the manuscript aligns with your requirements.
>
> Thank you for your time and support throughout this process. We truly appreciate your guidance and look forward to your comments.
>
> Best regards,
> Authors

---

### Review · Reviewer_RuQ8 · 2024-11-25

**Summary Of Contributions:**

The paper presents an analysis of independent sampling procedures  (ISP) vs random samplin procedures (RSP) for the selection of participating clients in federated learning. The former (ISP) being bernoulli samples with per-client probabilities and the latter being choose-k without replacement. The paper studies the contribution of the SP to the variance of the gradient estimate, focusing on unbiased client sampling, and shows that optimal independent sampling strictly outperforms RSP in terms of the sampling quality, accelerating the convergence rate by an almost linear factor. This is done by seperating the variance due to client heterogeneity from variance due to the estimation. Experiments are presented on synthetic, F-EMNIST and two NLP datasets with what used to be LLMs (pythia on CCnews/AGNews).

**Audience:**

Yes

**Broader Impact Concerns:**

I think no BIS is required.

**Claims And Evidence:**

Yes

**Requested Changes:**

The point about bias via hard capping is critical and would need to be addressed, the rest are nice to haves.

**Strengths And Weaknesses:**

# Strenghts
- I am reading a revised version, with various improvements. As such, the paper was relatively straightforward to follow through it's introduction
- As far as I can tell, relevant previous works are included
- For a theory paper, strong experimental validation with good results that follow the theory
- in particular I want to commend the authors for adding _interpretations_ to their prose (partially as a revision it appears, but nonethelss commendable)

# Weaknesses

## Nits

- on page 4, last paragraph,

>Interestingly, the ISP estimates perform worse than the uniform sampling baseline with a larger budget, as shown in the third plot of Figure 1(a). This discrepancy arises because the optimal probability in ISP aims to minimize a loose upper bound on variance, as defined in equation equation 3. This is because the optimal probability of RSP is minimizing a loose upper bound of variance equation 3

appears to refer to _RSP_ not ISP in the first two instance, and the sentence is doubled

- Interpretation of Theorem 4.1 paragram

>If we always use optimal client sampling (Chen et al., 2020), the cumulative sampling quality β1 = 0.

is probably meant to read

>If we always use optimal client sampling (Chen et al., 2020),  the cumulative sampling quality implies β1 = 0.

or something like it?

## More serious cirtique

- changing the definition of budget from strict K to probabilitic K implies that that per iteratoin, we might have much more or much less than K clients selected, unless I am mistaken? This means that the algorithm is mainly applicable if we are operating at a scale in which we are softly budget constrained, and it might be worth studying a setting in which RSP with hard $K_h$ is compared with ISP with a soft $K_s$ chosen to ensure that $P(|S_t| >K_h) \leq  \eps$  with $\eps \to 0. Alternatatively it could be commented on in the prose or the impact of being capped by a hard constraint would need to be analyzed (I would assume it would simply introuce introduce a bias term that needs to be controlled ?)

-  given the different training distributions on the real/synthetic dataset, I feel like a rebalancing/importance sampling strategy with fixed-but-nonuniform sampling probabilities would represent a stronger baseline, unless it is established that we assume we do not know the distribution across clients (I might have missed this but I think it wasn't assumed?)

## Not a critique but a curiosity

Both the $\gamma$ and the mixing incentivise the algorithm to hew close to the uniform probablity while allowing divergence. I would be interested in seing the deviatoi from the unfirom distribution (e.g. in wasserstein or TV ) across the optimization procedure, as it would shed some light in what particular types of adaptivity occur. The real world experiments indicate that when local client variance dominates the heterogeneity, the sampling

---

> ### Author Response · Authors · 2024-11-30
> **Thanks for your feedback!**
>
> We thank the reviewer for the valuable time and effort in reviewing our work. We also appreciate insightful comments and suggestions on our work. The following points are our response to the comments and all revisions are marked in blue.
>
> **Regarding Nits:** Thanks for pointing this out. We have revised it.
>
> **Regarding Serious critique:**
> **A1**: Thank you for the excellent suggestion, which helps clarify our work. It is correct that we might have more or less than $K$ clients selected by ISP. We studied the ISP with hard capping as required. It turns out to introduce the following new constraint on sampling probability：
> $$ \sum_{i=1}^N p_i(1-p_i) \geq -2(K_h-K_s)^2/\ln \epsilon.$$
> In revision, we provide a one-sentence discussion in the paragraph "ISP creates expected sampling size" in Section 3 with a detailed discussion in Appendix E.2 due to page limitation.
>
> **A2**: To ensure a fair comparison, all samplers in our experiments are unaware of the client data distribution and are initialized with uniform sampling probabilities. We believe this setting better matches the real-world FL practices, reflecting the necessity and significance of adaptive client sampling.
> However, we agreed that using initial probabilities based on prior knowledge (e.g., data distribution) to set initial sampling probabilities could enhance adaptive samplers. We plan to study in future work.
>
> **Regarding curiosity comments：**
> This comment suggested a good view. In response, we re-conducted the experiments and checked the TV distance across optimization procedures on NLP tasks and Synthetic tasks.  Here are our findings:
> - For all tasks, the TV distance increases over communication rounds, while the speed reduces over rounds. Large $\theta$ and small $\gamma$ allow the TV distance to be large, which means the sampling probabilities converge under the hyperparameter control.
> - We find that the trends of TV distances are related to data distribution across clients. The TV distances change more with a large degree of heterogeneity across clients. This indicates that K-Vib can accurately be adaptive to data distributions across clients.
> In revision, we add the TV deviation plot of probability in Figure 3 (c) and the TV deviation plot over communication rounds on NLP tasks in Figure 9, Appendix F.
>
> Thanks again for your insightful comments, and we believe these revisions have further strengthened our paper and enhanced clarity.
>
> **Kind reminder:** The last sentence in the review seems missing. We hope we didn't miss some important comments. Please let us know if any further revisions or concerns need to be addressed.

---

> > ### Comment · Reviewer_RuQ8 · 2024-12-01
> > **Thanks for adressing my concerns**
> >
> > Dear authors
> >
> > the sentence ended with something like "the sampling appears to offer less benefit", so the TV plot already incorporates that.
> >
> > I did some quick analysis and if I didn't make a mistake, the bound implies a condition of |K_h - K_s| \geq \sqrt{\frac{N \cdot \ln(1/\epsilon)}{8}} i.e., the "real cap" must be greater than the algorithmic soft cap by this margin, which is dominated by the sqrt(N) term. That is quite a nice property to have! Especially because presumably one could further mitigate the effect of oversampling.
> >
> > Overall, all my serious concerns have been addressed and I think the paper has again be strengthened, good job :-)

---

> > > ### Author Response · Authors · 2024-12-01
> > > **Thanks!**
> > >
> > > Thanks for your acknowledgment of our response and taking the time to review our paper!

---

### Author Response · Authors · 2024-12-05
**Summaries of revisions during the review period**

Dear reviewers,

We are grateful to all of you for taking the time to review our paper, for engaging in the subsequent discussions, as well as for providing constructive feedback. We believe the peer-review process has substantially improved the quality of our work. In what follows, we list the changes that we have made to the current manuscript:

- **Better comparison ISP v.s. RSP**: We provide a new Example 3.1 with a better discussion to clarify their differences.
- **Hard capping for ISP**: We provide a solution for the needs that further mitigate the effect of oversampling in Appendix E.2.
- **Better theoretical analysis:** We break up long chains of equations and provide an overview at the beginning of long proofs for better reading experiences.
- **New discussion and experiments on hyper-parameters:** we provide a clear discussion on the implications of hyperparameters at the end of section 5. And, we provide clearer sensitive study results on hyperparameters in Figure 3 (b),(c).
- **Probability variation:** We provide additional experiments in Figure 9 and discussion in Appendix F.2 to clarify the adaptive relation between probabilities and data distribution.
- **Better organization:** To provide more informative results in the main paper, we move the distribution plots of Synthetic and FEMNIST datasets to the appendix.
- **Revised minors:** We have fixed all mentioned minors and other potential minors.

Thanks for all the reviewers' comments, and we believe these revisions have greatly strengthened our work. As the discussion period is ending soon, we kindly ask if there are any remaining concerns or issues we could address or clarify.

Sincerely,

Authors

---

### Decision · Action_Editor_oXaK · 2025-01-06

**Recommendation:** Accept as is

**Comment:**

This paper proposes an adaptive sampling strategy for client selection in federated learning, guided by an online variance reduction technique.

Specifically, the paper:
- advocates for the arbitrary sampling framework (Horváth & Richtárik, 2019; Chen et al., 2020) for client sampling in federated learning,
- presents a convergence upper bound for FedAvg with arbitrary client sampling, highlighting the impact of the sampling quality measure on convergence,
- extends the online learning VRB (Borsos et al., 2018) to multi-client sampling (K-ViB) and establishes a regret bound for this procedure,
- evaluates the K-ViB sampler through numerical experiments.

This manuscript has been revised based on the feedback from reviewers across two rounds of review. The presentation has been substantially improved and now includes enhanced discussions and interpretations of the results, improved numerical experiments, and strengthened theoretical findings. All reviewers recommend acceptance of the manuscript.

**Audience:**

The topic is clearly of interest to parts of the TMLR community. The work proposes new client sampling strategies in federated learning.

**Claims And Evidence:**

The reviewers found the claims presented in the paper to be accurate and supported by sufficient evidence.

The experiments demonstrate that the proposed K-ViB outperforms the provided benchmarks. However, the reviewers noted that the set of comparisons is relatively limited, and thus, the claims might not generalize beyond the considered class of examples.